



# How can regime characteristics of catchments help in training of local and regional LSTM-based runoff models?

Reyhaneh Hashemi[1], Pierre Brigode[2,3], Pierre-André Garambois[1], and Pierre Javelle[1]

[1]Aix-Marseille Université, INRAE, UR RECOVER, Aix-en-Provence, France
[2]Université Côte d'Azur, Observatoire de la Côte d'Azur, CNRS, IRD, Géoazur, Sophia-Antipolis, France
[3]Université Paris-Saclay, INRAE, UR HYCAR, Antony, France

**Correspondence:** Reyhaneh Hashemi (reyhaneh.hashemi@inrae.fr)

**Abstract.** In the field of Deep Learning, the long short-term memory (LSTM) networks lie in the category of recurrent neural network (RNN) architectures. The distinctive capability of the LSTM is learning non linear long term dependency structures. This makes the LSTM a good candidate for prediction tasks in non linear time dependent systems such as prediction of runoff in a catchment. In this study, we use a large sample of 740 gauged catchments with very diverse hydro-geo-climatic conditions

across France. We present a regime classification based on three hydro-climatic indices to identify and classify catchments with similar hydrological behaviors. We do this because we aim to investigate how regime derived information can be used in training LSTM-based runoff models. The LSTM-based models that we investigate include local models trained on individual catchments as well as regional models trained on a group of catchments. In local training, for each regime, we identify the optimal lookback, i.e. the length of the sequence of past forcing data that the LSTM needs to work through. We then use

this length in training regional models that differ in two aspects: 1) hydrological homogeneity of the catchments used in their training, 2) configuration of the static attributes used in their inputs. We examine how each of these aspects contributes to learning of the LSTM in regional training. At every step of this study, we benchmark performances of the LSTM against a conceptual model (GR4J) on both train and unseen data. We show that the optimal lookback is regime dependent and homogeneity of the train catchments in regional training has a more significant contribution to learning of the LSTM than the

number of the train catchments.

## 1   Introduction

Surface runoff (in short, runoff) is the response of catchment to its intakes and yields. A reliable prediction of runoff is key information to many water related hazards and management of water resources and has been the focus of numerous studies in hydrology over the past several decades. Nevertheless, an accurate prediction of runoff has since remained a challenge due

to non linearity of the several involved surface and sub-surface processes (Kachroo and Natale, 1992; Phillips, 2003) and, in particular, parameter identifiability issues when it comes to ungauged catchments (Hrachowitz et al., 2013; Montanari et al., 2013; Blöschl et al., 2019).

Early studies of runoff prediction have used either conceptual models (Horton, 1933; Nash, 1957; Dooge, 1959; Betson, 1964; Bergstrom and Forsman, 1973) or physically based models (Abbott et al., 1986a, b; Refsgaard et al., 2010) with different





degrees of success. Later, neural network models of rainfall-runoff (R-R) process began to emerge and grow. Examples include models developed using artificial neural networks (ANNs), which were yet limited to short term runoff forecasting (Hsu et al., 1995; Minns and Hall, 1996; Anctil and Lauzon, 2004; Abrahart et al., 2012). Promising continuous data driven models have been presented only very recently. The runoff model developed by Kratzert et al. (2018) using long short-term memory (LSTM) networks (Hochreiter and Schmidhuber, 1997) was the first of its kind. It was a daily "many-to-one" model, i.e. it used

a sequence (therefore "many") of past forcing variables to predict a single (therefore "one") discharge value. The LSTM is a variant of recurrent neural networks (RNNs) and compared to ANNs or traditional RNNs has the main advantage of being able to deal with vanishing gradients. The vanishing gradient problem occurs when during model training back propagated gradients tend to zero due to computational issues related to finite precision numbers (Hochreiter, 1998; Goodfellow et al., 2016). Thanks to this property, the LSTM is able to capture long term time dependencies. This is advantageous to continuous R-R simulations

since transformation of rainfall to runoff depends hugely on the current anterior soil-water state of catchment, which is itself a result of antecedent meteorological conditions, sometimes going back to several months ago. This property of the LSTM and its successful application in Kratzert et al. (2018) has encouraged researchers since after to explore more widely the predictive capability of LSTM-based runoff models (Kratzert et al., 2019b; Gao et al., 2020; O et al., 2020; Frame et al., 2021; Gauch et al., 2021). Meanwhile, gated recurrent units (GRUs), which are a variant of LSTM networks (Cho et al., 2014), have been

also tested and compared against the LSTM for both prediction and forecasting applications and performances similar to those shown by the LSTM were obtained (Gao et al., 2020; Ayzel and Heistermann, 2021; Zhang et al., 2021).

In the previous studies by Kratzert et al. (2018) and Lees et al. (2021), the length of the input sequence, i.e. the LSTM's window size for looking to the past and hereafter called lookback, was set to 365 [day] so that the dynamics of a full annual cycle could be captured. However, Gauch et al. (2021) showed that a too long input sequence could impair performance when the available

data were limited and lookbacks shorter than 365 [day] should be favored. Kratzert et al. (2019b) tested four lookbacks (90, 180, 270 and 365 [day]) and reported that a lookback of 270 [day] gave the best results in their study.

One interesting aspect in the earlier study by Kratzert et al. (2018) was the implementation and comparison of "local" and "regional" LSTM-based models. In doing this, while the local LSTM used only data from one single catchment, the regional LSTM was trained based on the information from a group of catchments. This aspect has been further investigated in Kratzert

et al. (2019a, b) using 531 catchments of the CAMELS dataset (Addor et al., 2017) produced by the US National Center for Atmospheric Research (NCAR). Kratzert et al. (2019a, b) showed that the regional LSTM models that used both dynamic (e.g. forcing data) and static (e.g. catchment attributes) features outperformed the regional LSTM model that did not take into account any static features as well as all tested local benchmark models. In the climatic context of Great Britain and on a sample of 518 catchments, Lees et al. (2021) observed also an outperformance of regional LSTM models over four

conceptual benchmark models, namely SACRAMENTO, ARNOVIC, TOPMODEL and PRMS. These studies (Kratzert et al., 2018, 2019a, b; Lees et al., 2021) show that regional training of the LSTM brings performance improvement when compared with local training since regional models learn better. It is therefore tried to improve learning by increasing the number of train catchments. Kratzert et al. (2018) discuss that, in regional training, not only the train data is significantly augmented, but also different contributing catchments would bring different complementary information about R-R transformation under more





general hydrological conditions and, consequently, learning would improve.

Given the previous studies, we find two axes that we think can receive more attention. First, the question around the length of lookback that the LSTM "really needs" to reproduce runoff adequately. Second, the common assumption that increasing the number of train catchments in regional training will always lead to a better learning of the LSTM. This paper therefore moves towards a more hydrological investigation of these two axes and aims to explore how we can benefit from domain knowledge

in addressing them. In this perspective, this paper tries to answer the following two research questions:

1. In addition to the size of train data, does the optimal lookback depend on any hydrological factor? More specifically, is there a link between this "model memory" parameter and catchment "hydrological memory"?

2. In order to enhance learning in regional training, is it more effective to literally increase the number of train catchments of any hydrological class or, instead, to pool only hydrologically similar catchments? In other words, which aspect should

be favored in regional training, "hydrological homogeneity" or the number of train catchments?

To address these questions, this paper develops local and regional many-to-one daily LSTM models and apply them to a large sample of 740 French catchments with very diverse hydro-geo-climatic conditions. The models are benchmarked against the GR4J model, which is an acknowledged conceptual R-R model (Perrin et al., 2001).

The remainder of this paper is organized as follows. The following section presents the available data and proposes a catchment

classification reflecting the different hydrological regimes of the study catchments. Section 3 details the methods used in this work and describes the conducted experiments. Results are provided in Section 4 where the research questions are also discussed. The paper is concluded in Section 5, which also outlines some future perspectives based on the findings of this study.

## 2    Data

### 2.1    Hydroclimatic data and catchment attributes

The dataset used in this study contains time series of hydro-meteorological variables and time invariant catchment attributes. It is a subset of a larger dataset of 4190 French catchments processed and prepared by the HYCAR (Hydrosystemes Continentaux Anthropises-Ressources, Risques, Restauration) research unit of INRAE (Delaigue et al., 2020; Brigode et al., 2020).

The meteorological forcing data are produced by the daily SAFRAN (Système d'Analyse Fournissant des Renseignements Atmosphériques à la Neige) reanalysis that is run by Météo France at a resolution of $8 \times 8$ [km$^2$] (Quintana-Segui et al., 2008;

Vidal et al., 2010). For each catchment, spatially averaged forcing data consisting of daily total precipitation, mean air temperature, minimum air temperature, maximum air temperature, wind speed, air moisture, atmospheric radiation, and visible radiation, are available for the common period from 1958-08-01 through 2019-07-31.

The hydrometric data consist of daily time series of discharge and are collected by the French Ministry of Environment covering the period of the forcing data.

Our catchment sample includes 755 catchments from all over France having 21 to 60 [year] of full record data. These catchments vary in size from 1.8 to 96016 [km$^2$] with a median size of 192 [km$^2$]. The annual runoff ranges from 25 to 2312





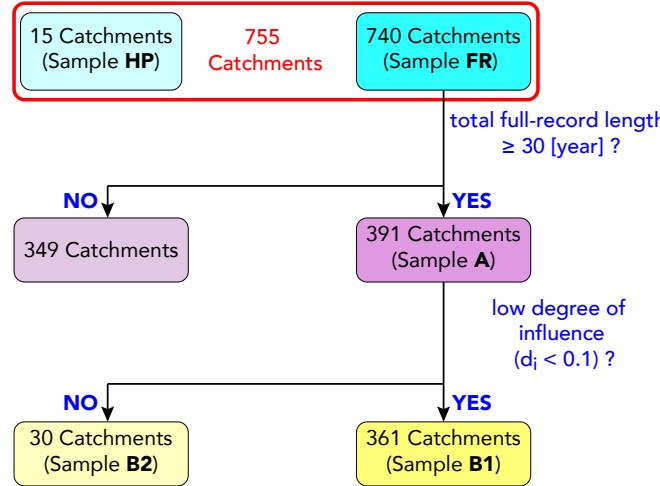

**Figure 1.** Different catchment samples used in different experiments of the study

[mm per year], with a median value of 444 [mm per year] and the annual total precipitations varies between 563 and 2290 [mm per year], with a median value of 1019 [mm per year]. The mean daily temperature of the catchments varies between -1.8 and 14.8 [°C] and has a median value of 9.5 [°C].

We define influenced catchments as those with identified dam(s) within their topographic area upstream of their measuring station for which the volume of water storage by dam reservoir(s) is not null. For each catchment in our sample, we have the information on the reservoir storage volume [$m^3 \times 10^6$] from Brigode et al. (2020). To represent the degree of influence ($d_i$), we divide this volume by the surface area of catchment [$km^2$] to account for the fact that a given anthropogenic water intake volume does not have the same impact on a small catchment as on a large catchment. Catchments with a $d_i$ value equal to 0

are put aside as non influenced. For the remaining catchments, the tertiles, partitioning non zero $d_i$ values into 3 sub-groups of (nearly) equal sizes, are calculated. Catchments in the last tertile ($d_i \geq 0.1$) are considered to be catchments with a high degree of influence.

From the 755 catchment sample, we use different subsets in different experiments of this paper. These subsets differing in length of full record data and $d_i$ are presented in Fig. 1.

**2.2  Catchment classification**

The classification used here is inspired by Pardé (1933) and Sauquet (2006) and is based on three interannual monthly variables, namely runoff ($Q$ [mm per year]), total precipitation ($P_{\mathrm{tot}}$ [mm per year]), and minimum temperature ($T_{\mathrm{min}}$ [°C]). From





these variables, the three classification indices $IQ$ [-], $IP$ [-], and $T_{\min}$ [°C] are defined as follows for each catchment:

$$IQ = \frac{Q_{\max} - Q_{\min}}{Q_{\mathrm{mean}}} \tag{1}$$

$$IP = \frac{P_{\max} - P_{\min}}{P_{\mathrm{mean}}} \tag{2}$$

$$T_{\min} = \min(T_1, ..., T_i) \quad i \in 1, 2, ..., 12 \tag{3}$$

where $T_i$ is mean annual temperature of month $i$. In this definition, the $IQ$ and $IP$ indices give information on runoff variability and precipitation variability over the year, respectively. Low values of $IQ$ and $IP$ indicate a uniform distribution of them over the year, while a high value indicates the presence of contrasted dry and humid seasons. A low $IQ$ can also imply

the presence of ground water effects or reservoirs (natural or artificial) tending to attenuate runoff fluctuations at catchment outlet. The $T_{\min}$ index is a proxy to determine whether or not precipitation is received as snow during winter time. Figure 2 shows spatial variations of the three indices within the whole sample. High $IQ$ levels are fragmented in patches in the east, west, and south of France. The wettest areas are found on the Mediterranean coast in the south of France. $T_{\min}$ is lowest in the mountainous areas: the Alps in the east, the Pyrenees in the south west, and the Massif Central in the center of France.

Based on the defined indices ($IQ$, $IP$, and $T_{\min}$), the following classification criteria are applied to all catchments in the sample to determine their hydrological regime (Fig. 3):

Nival:           $T_{\min} < -2$

Nival-Pluvial:    $-2 < T_{\min} < 0$

Mediterranean:  $T_{\min} \geq 0$ and $IP > 1$

Uniform:        $T_{\min} \geq 0$ and $IP \leq 1$ and $IQ < 1$

Oceanic:        $T_{\min} \geq 0$ and $IP \leq 1$ and $IQ \geq 1$

The location of catchments within each regime is shown in Fig. 4. It is observed that the regimes are geographically plausible and compatible with region geographic characteristics. For example, the Nival and Nival-Pluvial regimes occur in the moun-

tainous ranges and the catchments of the Mediterranean regime are found along the French Mediterranean coastline and in the Mediterranean Corsica island. The oceanic catchments are spread across other parts of France, except areas known to have important aquifers which are held by the catchments of the Uniform regime (e.g. the Paris Basin region in the north of France). For each regime, variations of interannual monthly runoff, total precipitation, and mean temperature over the year is presented in Fig. 5. In the Uniform regime, runoff remains in the range between 6% and 10% of the annual discharge all over the year and

no wet or dry period is observed. This is while the other regimes clearly exhibit periods of low and high flows. The Oceanic regime is characterized by low flows during the summer and high flows during the winter. This is due to higher evaporations in summer with respect to winter. Total precipitation displays a rather uniform pattern in this regime. For catchments in the Mediterranean regime high flows have a wider period but are less pronounced compared to the Oceanic regime. However, low flows occur at lower levels as a result of extremely dry summers. Autumn precipitation is abundant in this regime, making

autumn a period prone to thunderstorms which could in turn induce sudden flash floods. Pattern of runoff in the Nival class is also recognizable with its snowmelt induced peak in the late spring/early summer where there is a rise in temperature. The





Nival-Pluvial regime appears to be a combined regime of the Oceanic and Nival regimes with two high flow periods, in autumn and spring.

## 2.3 Catchment physical attributes

The quartile distribution of catchment surface area [km$^2$], median slope [%], median drainage density [%], and median altitude [m] within different regimes is shown in Fig. 6. It is seen that surface areas in all regimes are spread between the four quartiles. That is, all regimes have catchments from almost all four quartiles. This is not however the case for other attributes. For instance, catchments having the highest 25% of values of altitude or slope are more likely to belong to the Nival or Nival-Pluvial regimes. Similarly, it is more probable that catchments with the lowest 25% of drainage densities belong to the Uniform regime than to

the Nival or Mediterranean regimes.

## 3 Methods

### 3.1 Principles of the LTSM

The LSTM is a subclass of RNN algorithms and has proven well suited to modeling a time dependent system where there can be unknown lags between the response of the system to a continuous input to it. This is the case for transformation

of meteorological signal into runoff in a catchment. In the context of Deep Learning, capturing time dependencies can be translated as "sharing important information between time steps of a time sequence" (Goodfellow et al., 2016). Information sharing of RNNs is designed to be very deep. However, in practice, this is not the case for very long time sequences due to the vanishing gradient problem. The LSTM is designed in turn to allow both shallow and deep information sharing across a sequence. In an ordinary RNN cell information sharing is through a feedback connection ($h_t$, also called "hidden state")

that maintains a shallow flowing memory of states of the sequence. The hidden state is therefore able to capture short term time dependencies well. An LSTM cell involves an extra state feedback connection ($C_t$, also called "cell state") that runs in parallel to the hidden state and retains information between two points, which can be very far away in the sequence (long term memory). The information exchange with the cell state is regulated via three internal gates (Fig. 7). The gates are neural networks (NNs) in nature and involve weights, biases, and sigmoid activation functions. The sigmoid activation functions

output values between zero (to completely throw away information) and one (to fully retain information). At each time step $t$, first the forget gate $f_t$ decides what information from the previous time step ($t-1$) is important to maintain in the cell state (Equation 4). The input gate $i_t$ (Equation 5) together with a tanh layer (Equation 6) decides what new information from the current time step is passed through the cell state. At this stage the cell state gets updated (Equation 7). Next, the output gate $o_t$ determines what information is output from the cell state (Equation 8). Finally, a tanh layer updates the hidden state (Equation





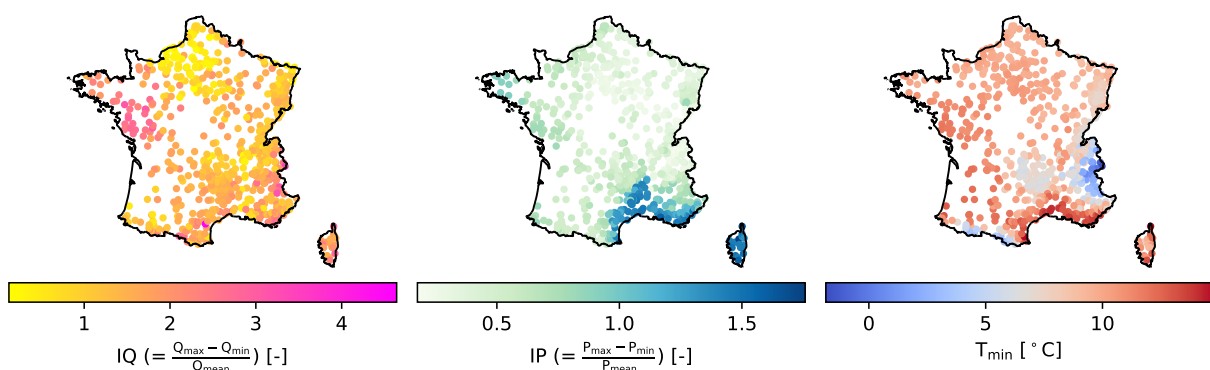

**Figure 2.** Spatial variation of the three regime classification indices $IQ$, $IP$, and $T_{\min}$ within the HP and FR samples.

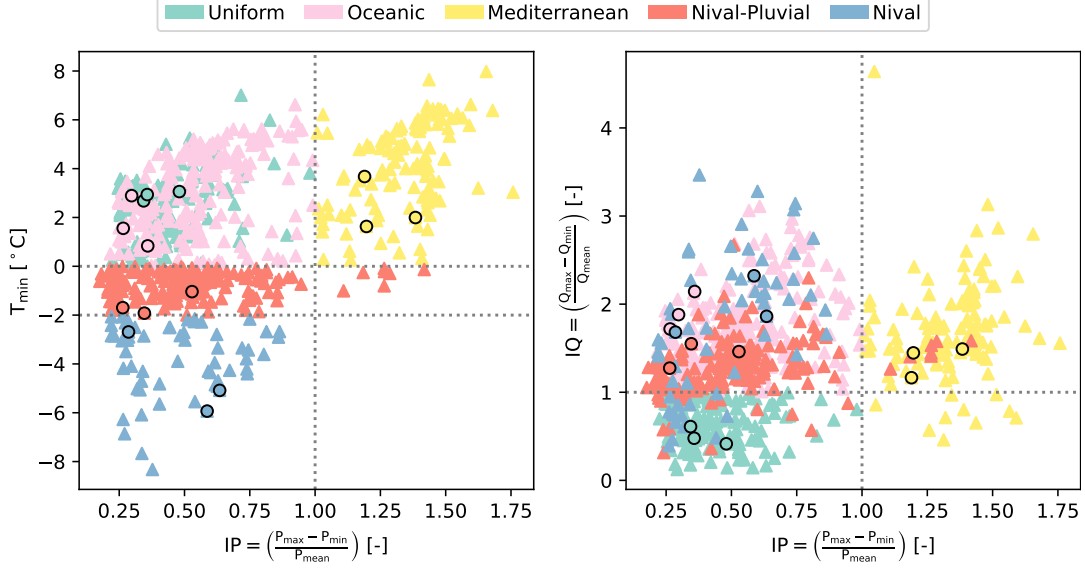

**Figure 3.** Classification of the catchments into five hydrological regimes based on the indices $T_{\min}$, $IP$, and $IQ$, in order of priority. Points represent catchments. Symbol ($\bigcirc$) represents catchments within the HP sample. The other points correspond to the catchments in the FR sample

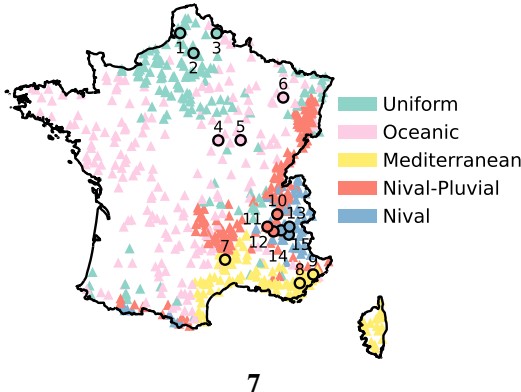

**Figure 4.** Location of the catchments within each regime. Points represent catchments. Symbol ($\bigcirc$) represents catchments within the HP sample. The other points correspond to the catchments in the FR sample



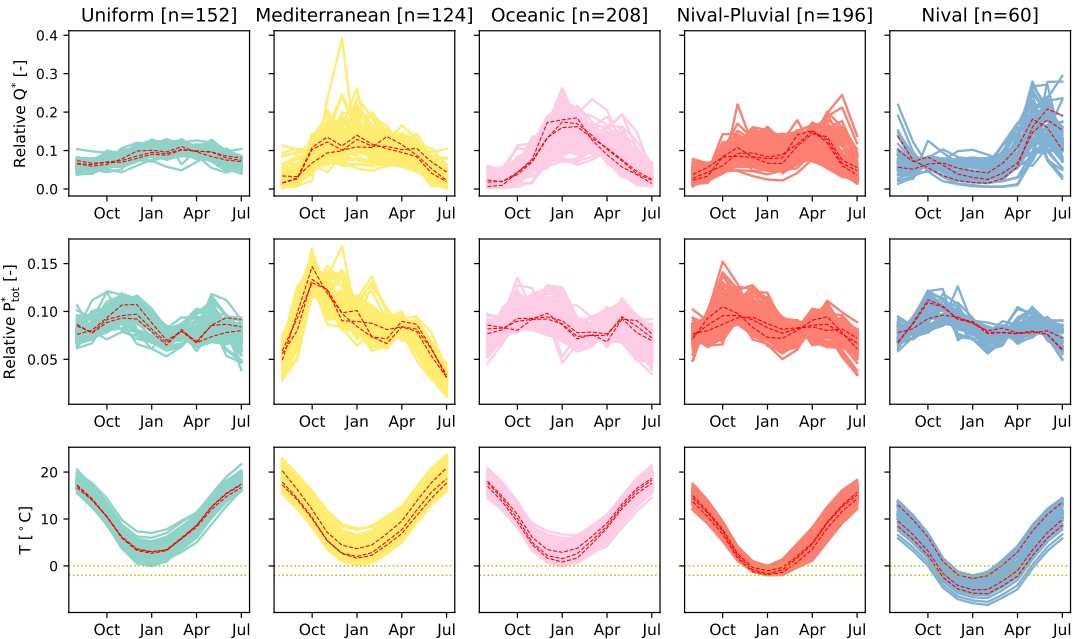

**Figure 5.** Interannual monthly runoff ($Q^*$), total precipitation ($P^*_{tot}$), and temperature ($T$) for the catchments in the FR sample (solid lines) and the catchments in the HP sample (red dotted lines). The ($^*$) symbol in $Q^*$ and $P^*_{tot}$ indicates that values of these two variables are relative to the total annual amount.

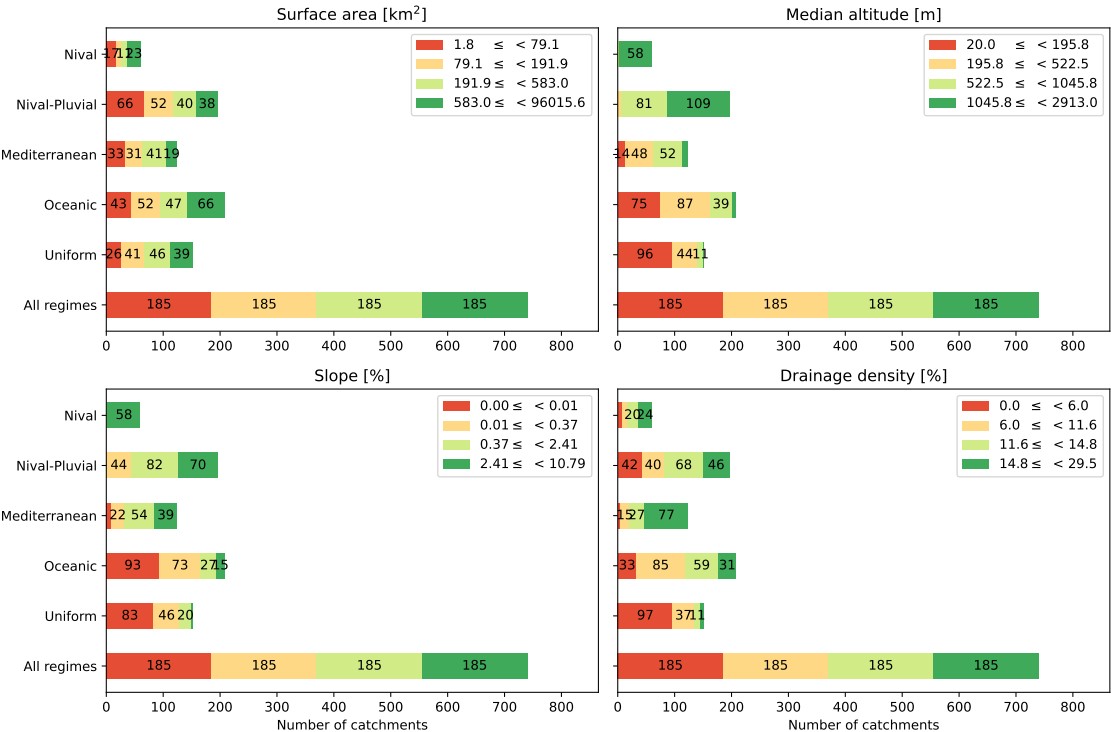

**Figure 6.** Stacked bar charts depicting variation of physical attributes within the regimes of the FR sample. The end-to-end segments of each bar correspond to the intervals delimited by the quartiles of the variable of interest. The quartiles are computed taking all catchments into account. Numbers show segment lengths and, due to space limits, are not displayed for values equal to or smaller than 10.





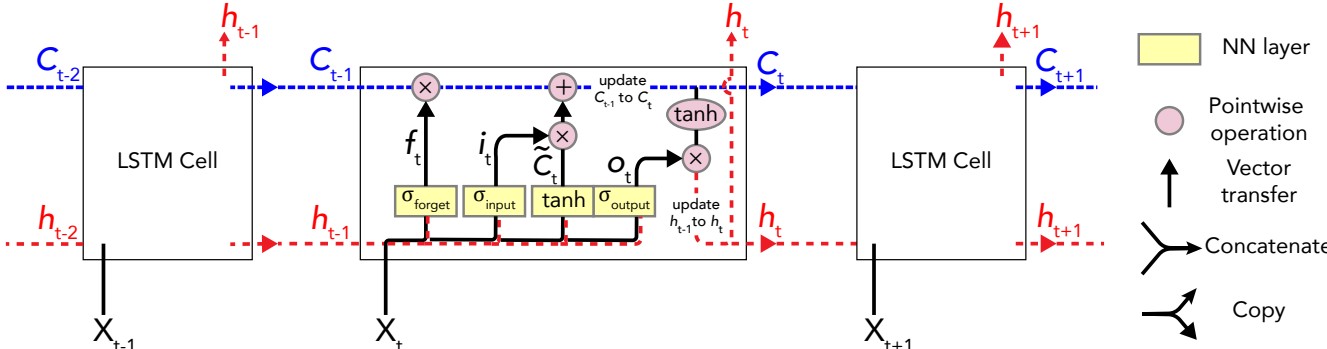

**Figure 7.** Time unfolded schematic representation of data processing through successive LSTM units. $X_t$ is the input of time step $t$. $\boldsymbol{h_t}$ is the hidden state (dashed red line). $\boldsymbol{C_t}$ represents the cell state (dashed blue line). $\sigma$ and $\tanh$ are activation functions. NN stands for neural network. The figure is adapted from Chollet and Allaire (2018) and Olah (2015).

9). For a detailed description of LSTM networks, please refer to Goodfellow et al. (2016).

$$\boldsymbol{f}_t = \sigma\left(\mathbf{W}_\mathrm{f}[X_t,\, \boldsymbol{h}_{t-1}] + \boldsymbol{b}_\mathrm{f}\right) \tag{4}$$

$$\boldsymbol{i}_t = \sigma\left(\mathbf{W}_\mathrm{i}[X_t,\, \boldsymbol{h}_{t-1}] + \boldsymbol{b}_\mathrm{i}\right) \tag{5}$$

$$\tilde{C}_t = \tanh\left(\mathbf{W}_\mathrm{C}[X_t,\, \boldsymbol{h}_{t-1}] + \boldsymbol{b}_\mathrm{C}\right) \tag{6}$$

$$\boldsymbol{C}_t = \boldsymbol{f}_t \times \boldsymbol{C}_{t-1} + \boldsymbol{i}_t \times \tilde{\boldsymbol{C}}_t \tag{7}$$

$$\boldsymbol{o}_t = \sigma\left(\mathbf{W}_\mathrm{o}[X_t,\, \boldsymbol{h}_{t-1}] + \boldsymbol{b}_\mathrm{o}\right) \tag{8}$$

$$\boldsymbol{h}_t = \boldsymbol{o}_t \times \tanh(\boldsymbol{C}_t) \tag{9}$$

Where $X_t$ is the input of the hidden layer. $\boldsymbol{f}_t$, $\boldsymbol{i}_t$, and $\boldsymbol{o}_t$, represent the forget, input, and output gates of LSTM cells, respectively. The notation $[X_t,\, \boldsymbol{h}_{t-1}]$ is used for vector concatenation between $X_t$ and $\boldsymbol{h}_{t-1}$. $\sigma$ and $\tanh$ are the sigmoid and hyperbolic tangent functions. $\mathbf{W}_f$, $\mathbf{W}_\mathrm{i}$, $\mathbf{W}_\mathrm{C}$, $\mathbf{W}_\mathrm{o}$ are learnable weight matrices. $\boldsymbol{b}_\mathrm{f}$, $\boldsymbol{b}_\mathrm{i}$, $\boldsymbol{b}_\mathrm{c}$, and $\boldsymbol{b}_\mathrm{o}$ are learnable bias vectors.

## 3.2   Application of the LSTM to runoff prediction

It is intended to use an LSTM model that takes the input data and predicts the observed daily runoff as target ($y_t$ [mm per day]) for time step $t$ ($\in 1, 2, \ldots, T$) within a given catchment. The input data consist of $N$ time varying forcing variables ($X_t$), also called dynamic features: $X_t = (X_{1,t}, \ldots, X_{N,t})$. In regional training, the input data include also $M$ time invariant variables (such as climatic and physical attributes), also called static features: $a = (a_1, \ldots, a_M)$. For this catchment and time step,

the predicted runoff ($\widehat{y_t}$) is the output of function $\mathcal{P}$ mapping the $L$-element input sequence $(X_{t-L+1}, \ldots, X_t)$ (which will be $([X_{t-L+1}, \ldots, X_t, a])$ in regional training) to the target at $L$ connected LSTM cells through the set of transformations





explained above (Fig. 7). Mathematically, this many-to-one mapping can be written as follows:

$$\mathcal{P}_\Omega : (X_{t-L+1}, \, \dots \, , X_t) \longrightarrow \widehat{y_t} \qquad\qquad \text{local training} \qquad\qquad (10)$$

$$\mathcal{P}_\Omega : ([X_{t-L+1}, \, \dots \, , X_t, a]) \longrightarrow \widehat{y_t} \qquad\qquad \text{regional training} \qquad\qquad (11)$$

Where $L$ is lookback. A loss function ($\ell$) then measures the prediction accuracy, i.e. how far the predicted runoff ($\hat{y}_t$) is from the observed runoff ($y_t$). The job is therefore to learn $\Omega$ (the best set of parameters of function $\mathcal{P}$) so that the loss function is globally minimized: $\underset{\Omega}{\arg\min} \left(\ell(\widehat{y}, y)\right)$.

In the case that static features are taken into account, $X_t$ in Equations 4, 5, 6, and 8 will be replaced by $[X_t, a]$, which is the concatenated vector of dynamic and static features. This implies that the static features are simply repeated at each time step.

## 3.3 Data split and standardization

A proper implementation of a Deep Learning model requires three independent subsets of data: train, validation, and evaluation sets. The train set is used to train the model (find model weights and biases). The validation set is intended to be used for finding the best weights and biases during training and control overfitting. The evaluation set provides independent data to evaluate the real model performance. Note that the fully independent validation and evaluation sets are both necessary in order to prevent

data leakage from the validation set to the model. This is because, although the information in the validation set is not directly used in model training, it is indirectly used to find the optimal model and prevent overfitting and should not be used to evaluate model performance.

In this study, since the period of full record discharge differs between catchments, the input data of each catchment are split according to its own full record period. To obtain the three sets, the first period from the end of time series that contains 10

years of full record discharge is set as the evaluation period (P3). The next subsequent period that contains 10 years of full record discharge is set as validation period (P2). What remains constitutes the train period (P1) the length of which varies between 1 year to 40 years in the FR sample.

The features and target in our dataset vary in wide ranges of values. This could decelerate or destabilize the learning process or make it fail as a result of back propagated gradients tending to infinity. To avoid such issues, we perform a feature wise

standardization of the features and the target data. The standardization is performed based on the mean and the standard deviation of the train set.

## 3.4 Performance criterion

To evaluate prediction performance, we use Kling–Gupta Efficiency (KGE) measure (Gupta et al., 2009) since it combines the three fundamental diagnostic properties of a predictive hydrological model, i.e. variability ($\alpha$), bias ($\beta$), and linear correlation





$(r)$.

$$\text{KGE} = 1 - \sqrt{(1-\alpha)^2 + (1-\beta)^2 + (1-r)^2} \tag{12}$$

$$\alpha = \frac{\text{std}(\widehat{y})}{\text{std}(y)} \tag{13}$$

$$\beta = \frac{\overline{\overline{y}}}{\overline{y}} \tag{14}$$

$$r = \frac{\sum_{t=1}^{t=T}(y_t - \overline{y})\left(\widehat{y}_t - \overline{\widehat{y}}\right)}{\text{std}(\widehat{y}) \times \text{std}(y)} \tag{15}$$

Where std is the standard deviation function and $\overline{\widehat{y}}$ and $\overline{y}$ are mean values of $\widehat{y}$ and $y$.

### 3.5 Model setup

We use the Keras library (Chollet et al., 2015) of Python 3.8 (Van Rossum and Drake, 2009) to build the LSTM-based models. Depending on whether training is local or regional, the mean squared error function (MSE, Equation 16) or the NSE[*] (Equation 17, Kratzert et al. (2019b)) is used as the loss function, respectively. The NSE[*] is basin specific and is in particular useful in

regional training where information comes from different catchments and their discharge variance could vary in a wide range. The NSE[*] is normalized with respect to variance of discharge in each catchment. This will prevent giving smaller or larger weights to catchments with lower or higher variance.

$$\text{MSE} = \frac{1}{T}\sum_{t=1}^{T}(y_t - \widehat{y}_t)^2 \tag{16}$$

$$\text{NSE}^* = \frac{1}{K}\sum_{k=1}^{K}\sum_{t=1}^{T}\frac{(y_t - \widehat{y}_t)^2}{(s(k) + \epsilon)^2} \tag{17}$$

Where $K$ is the number of catchments, $T$ is the number of time steps for catchment $k$. $s(k)$ is the standard deviation of discharge for catchment $k$ and is computed using discharges of the train period (P1).

The gradient based Adam algorithm (Kingma and Ba, 2017) with a learning rate of 0.001 is used for optimization of the $\mathcal{P}$ function. The optimization algorithm is iterative taking place over a finite number of steps, where at each step the internal

model parameters (weights $\mathbf{W}$ and biases $\boldsymbol{b}$) are slightly improved resulting in a slightly lower global loss. The optimization at each step runs on a mini batch of data, i.e. a number of $L$-element sequences of data. "Batch size" is defined as the number of sequences (of size $L$) to deal with at each step before updating model weights and biases.

Train data are extracted in mini batches and are passed to the optimization algorithm. One passage of the entire train data through the optimization loop is called one "epoch". "Number of epochs" is defined as the number of times that the optimization

algorithm runs through the entire train data.

To control overfitting, we use the early stopping criterion of Keras. The early stopping algorithm tracks the loss value of the validation set and if it does not improve for $p$ successive epochs, the early stopping algorithm stops training and returns the



weights and biases giving the best results (minimum loss on the validation set). The parameter $p$ is called "patience" of the early stopping algorithm. If the loss continues to improve and the early stopping algorithm does not stop training, it will continue

until a "maximum number of epochs" ($N_{\mathrm{EP,max}}$) is reached.

To choose the number of LSTM layers and nodes, batch size, and lookback, we conduct a systematic set of preliminary experiments (EXP0) that will be presented in the following subsection. We use different values of parameters patience and maximum number of epochs in different experiments. These values will be specified in the corresponding experiment section.

### 3.6    Experiments

### 3.6.1    Hyperparameter tuning (EXP0)

Hyperparameters of a Deep Learning model are different from its internal learnable parameters (weights and biases) and must be tuned outside of learning process. For this purpose, the preliminary experiments EXP0 are conducted in which two architectures along with different batch sizes and lookbacks are tested. To reduce the hyperparameter tuning task to manageable proportions, we use the HP sample (Fig. 1). It includes only 15 catchments (3 catchments per hydrological regime) having

sufficiently long records ($\geq 40$ [year]) and not being influenced by dams ($d_i = 0$). The location and regime of these catchments have been presented previously in Fig. 4 and Fig. 5, respectively.

In EXP0, two architectures are tested. The first (S1) involves a single standard LSTM layer of 64 nodes (Lees et al., 2021) and the second (S2) consists of two standard LSTM layers of 32 nodes. Batch size is tested for values of 16, 32, 64, 128, and 256 and choices of lookback include 30, 60, 90, 180, 365, 730 [day].

Features in this experimental set include the following seven time varying variables at a daily time step: total precipitation ($P_{\mathrm{d,tot}}$), minimum air temperature ($T_{\mathrm{d,min}}$), maximum air temperature ($T_{\mathrm{d,max}}$), wind speed ($S_{\mathrm{d,w}}$), air moisture ($H_{\mathrm{d,a}}$), atmospheric radiation ($r_{\mathrm{d,a}}$), and visible radiation ($r_{\mathrm{d,v}}$).

Model training is performed locally and the MSE is used as the objective function. Patience and maximum number of epochs are set to 50 and 500 [epoch], respectively, to make sure that the model has enough time to reach its optimal point.

The KGE score of the P3 period in different batch size-lookback combination tests for the S1 and S2 structures are shown in Appendix A. It can be observed that performances depend more on lookback than on batch size. However, for most catchments, the KGE scores tend to drop when using a batch size larger than 128, most probably, since it would lead to poorer generalization. Therefore, to keep a reasonable tradeoff between number of required iterations (smaller for larger batch sizes) and model accuracy/generalization (higher for smaller batch sizes), we decided to set batch size to 128 hereafter. Comparing the

maximum achieved performance for different catchments of the two architectures, we can observe that the performance gain by using the more complex architecture S2 is slight, if any. Besides, the architecture S2 takes longer to train. For these reasons, we selected to use the simpler architecture S1 hereafter. Taking into account lookback, two observations can be made. First, for each catchment there is an optimal lookback above which no further performance improvements can be gained. Second, the optimal lookback appears to depend on the hydrological regime of catchment. Nevertheless, since the number of catchments

in the HP sample is too few in total (also, per hydrological regime), we decided to not to fix lookback at this stage and keep all





lookback sizes for the following experiment (EXP1). This would give the opportunity to validate these observations in a larger sample, which would be less susceptible to bias.

### 3.6.2  Local training (EXP1)

The EXP1 experiments are based on local training and seek to answer the first research question, i.e. whether the optimal
lookback can be linked to catchment regime. To perform EXP1, the FR sample is used (Fig. 1). The EXP1 series is conducted using the hyperparameters concluded from EXP0: the S1 architecture, a batch size of 128, and lookback values of 30, 60, 90, 180, 365, 730 [day]. Model training is performed locally (catchment by catchment) using the MSE as the loss function. The features in EXP1 consist only of the seven daily time varying (dynamic) variables as EXP0: total precipitation ($P_{d,tot}$), minimum air temperature ($T_{d,min}$), maximum air temperature ($T_{d,max}$), wind speed ($S_{d,w}$), air moisture ($H_{d,a}$), atmospheric
radiation ($r_{d,a}$), and visible radiation ($r_{d,v}$). Similar to EXP0, patience and maximum number of epochs are set to 50 and 500 [epoch], respectively.

### 3.6.3  Regional training (EXP2)

The EXP2 experiments are based on regional training and its conceptual framework is guided by the second research question. Here, we are interested in the performance improvement that hydrological similarities of training catchments could bring.
EXP2 is conducted using the B1 sample (Fig. 1). In EXP2, catchments with high degrees of influences ($d_i \geq 0.1$) are excluded to not trouble regional learning by catchments having a non natural hydrological behavior. EXP2 consists of four series based on two regional training approaches and two configurations of static features. In the first training approach, one model is trained using the whole B1 sample from whatever hydrological regimes. In the second approach, one model is trained per hydrological regime. In this approach, training is based on a fewer number of catchments but train catchments belong to the
same hydrological regime. Taking into account static features, two configurations are considered. In the first configuration, only physical catchment attributes (surface area ($A$ [km$^2$]), slope ($S$ [%]) and drainage density ($DD$ [%]) are used. In the second configuration, hydro-climatic attributes ($IQ$, $IP$, $T_{min}$, as defined in Section Data), are added to the physical features. Table 1 gives a summary of the training approach, sample, dynamic and static features, as well as patience and maximum number of epochs used in different experiments of this study. Note that patience and maximum number of epochs are decreased
in EXP2 since the available data during each training epoch of EXP2 are significantly larger compared to EXP0 and EXP1. Model training is carried out using specific lookbacks concluded from the results obtained in EXP1, which will be presented in the Results Section.

### 3.7  Benchmark model

We select the daily lumped GR4J model (Génie Rural à 4 paramètres Journalier (Perrin et al., 2003)) and its snowmelt routine
CemaNeige (Valéry et al., 2014) to benchmark the LSTM. Among conceptual models, GR4J is a proven one (Demirel et al., 2013; Broderick et al., 2016) and since its initial presentation by Michel (1989), it has been continuously improved and gained



**Table 1.** Description of different LSTM-based models investigated in this study

| Experiment | Training approach | Sample | Daily dynamic features | Static features | Patience | $N_{EP,max}$ |
|---|---|---|---|---|---|---|
| **EXP0** | one model per catchment (local) | HP | $P_{d,tot}, T_{d,min}, T_{d,max}, V_{d,w}, H_{d,a}, r_{d,a}, r_{d,v}$ | — | 50 | 500 |
| **EXP1** | one model per catchment (local) | FR | $P_{d,tot}, T_{d,min}, T_{d,max}, V_{d,w}, H_{d,a}, r_{d,a}, r_{d,v}$ | — | 50 | 500 |
| **EXP2 - R1** | one model for sample (regional) | B1 | $P_{d,tot}, T_{d,min}, T_{d,max}, V_{d,w}, H_{d,a}, r_{d,a}, r_{d,v}$ | $A, S, DD$ | 5 | 20 |
| **EXP2 - R2** | one model for sample (regional) | B1 | $P_{d,tot}, T_{d,min}, T_{d,max}, V_{d,w}, H_{d,a}, r_{d,a}, r_{d,v}$ | $A, S, DD, IQ, IP, T_{min}$ | 5 | 20 |
| **EXP2 - R3** | one model per hydrological regime (regional) | B1 | $P_{d,tot}, T_{d,min}, T_{d,max}, V_{d,w}, H_{d,a}, r_{d,a}, r_{d,v}$ | $A, S, DD$ | 10 | 50 |
| **EXP2 - R4** | one model per hydrological regime (regional) | B1 | $P_{d,tot}, T_{d,min}, T_{d,max}, V_{d,w}, H_{d,a}, r_{d,a}, r_{d,v}$ | $A, S, DD, IQ, IP, T_{min}$ | 10 | 50 |

$N_{EP,max}$: maximum number of epochs

The $d$ subscript indicates the daily resolution of variables.

$P_{d,tot}$: total precipitation, $T_{d,min}$ and $T_{d,max}$: minimum and maximum air temperature, $V_{d,w}$: wind speed, $H_{d,a}$: air moisture, $r_{d,a}$: atmospheric radiation, $r_{d,v}$: visible radiation

$A$: area, $S$: slope, $DD$: drainage density

recognition among researchers (Broderick et al., 2016; Wijayarathne and Coulibaly, 2020; Guo et al., 2020), engineers (Braccia et al., 2014), and operational authorities (Pagano et al., 2010), for scientific, educational, and practical purposes in France and worldwide (Pagano et al., 2010; Demetriades, 2020; Yang et al., 2020). This is because, while GR4J is a relatively simple

model to setup and requires few inputs, it can adequately represent the hydrological response of catchment (Perrin et al., 2003; Velázquez et al., 2010; Mathevet et al., 2020). Furthermore, GR4J is a parsimonious model incorporating only four free parameters and is able to deal with groundwater exchanges with aquifers and/or adjoining catchments thanks to a gain/loss function. This is a distinctive aspect of GR4J compared against the benchmarks models used in previous studies (Kratzert et al., 2018; Lees et al., 2021). CemaNeige is a conceptual snowmelt routine with two parameters and computes snow accumulation

and snow melt as outputs (Valéry et al., 2014).

Compulsory inputs to the GR4J model consist of daily total precipitation [mm/day], potential evapotranspiration [mm/day], and runoff [mm/day] where runoff is only used for model calibration. Compulsory inputs to the CemaNeige snowmelt routine are daily total precipitation [mm/day] and mean air temperature [°C]. We also used hypsometric data of each catchment as an optional input in the CemaNeige model. It uses this information to account for orographic gradients (Valéry et al., 2014). Note

that the inputs used in the GR4J+CemaNeige model are not the same as those used for the LSTM (Table 1). The optimization algorithm Michel (Michel, 1989) is used for model calibration. For the sake of comparison with the LSTM model, NSE is selected as objective function of the optimization algorithm. We perform two series of GR4J+CemaNeige simulations for each catchment. The first (GR4J1) consists of calibrating the model on P1 and evaluating it on P3. In the second series (GR4J2), the model is calibrated on P1+P2 and is evaluated on P3. In GR4J, it is recommended to take into account a period of warm-up to

provide the model with an initial state rather than starting with an arbitrary state (Perrin and Littlewood, 2000). Therefore, in both series and for both calibration and evaluation phases, we set the first 2 years of period as the warm-up period. The length of the warm-up period corresponds to the longest lookback tested for the LSTM. GR4J simulations are performed using the airGR package (Coron et al., 2017, 2020) in the R (R Core Team, 2019) interface. Note that all GR4J simulations in this study are performed at local scale and no regional simulation is carried out for the benchmark model.





## 4  Results

### 4.1  Local training (EXP1)

Figure 8 compares the KGE scores in EXP1 calculated on the P1+P2 and P3 periods for the LSTM and the GR4J models. The results are presented in four panels. Partitioning of the results is based on the 4 quartiles of the length of the P1+P2 periods of the catchments in the experiment sample (i.e. sample FR). Each panel shows the results of the catchments lying in one of the quartiles. For the LSTM, no matter the lookback size, increasing the length of the train data improves consistently the median KGE score on unseen data (P3). The best performances are therefore observed for catchments with the highest 25% of the train lengths (the lower right panel) and catchments with the lowest 25% of the train lengths show the worst performances (upper left panel). A similar, but to a lower extent, observation can be made for performances on the P1+P2 period. This indicates that increasing the train data brings performance improvement to the local LSTM in both terms of learning (performances on P1+P2) and generalization (performances on P3). However, data scarcity causes a major degradation of, not learning, but the generalization ability of the local LSTM.

In order to exclude data scarcity from possible sources of performance decline for the LSTM and to be able to study other factors independently, the following analysis at local scale is performed based on the results obtained from catchments having a train length of 30 years or more (the A sample, Fig. 1). The KGE scores of these catchments, taking them all into account and within each hydrological regime, are shown in Fig. 9.

The catchments of the A sample are distributed as follows in the five regimes: 78 catchments in the Uniform regime, 103 catchments in the Oceanic regime, 64 catchments in the Mediterranean regime, 110 catchments in the Nival-Pluvial regime, and 36 catchments in the Nival regime. Taking first all 391 catchments into account (the upper left panel of Fig. 9), the median KGE score continues to increase with increasing lookback until it becomes "stagnant" at 180 [day]. This indicates that long lookbacks do not necessarily lead to better performances. If now performances within different hydrological regimes are considered, except in the two contrasting Mediterranean and Uniform regimes, a similar pattern at lookback 180 [day] occurs. Within the Mediterranean regime, the major performance improvement occurs in very early stages of lookback increase and from 60 [day] on no further considerable improvement is observed. On the contrary, in the Uniform regime, the median KGE score is constantly rising as lookback increases up to 730 [day]. Taken together, in the Oceanic, Nival-Pluvial and Nival regimes, a lookback of 180 [day] and in the Mediterranean regime a lookback of, as short as, 60 [day] turns out to be sufficient to adequately reproduce runoff. However, in the Uniform regime, the LSTM can need to work through a lookback as long as 730 [day].

### 4.2  Regional training (EXP2)

The regional LSTM models in EXP2 include four models: R1, R2, R3, and R4. These models are trained based on the lookbacks that are concluded optimal at local scale (in EXP1). In the R1 and R2 models all catchments (of the B1 sample) are trained together and lookback is set to 180 [day]. In the R3 and R4 models one regional model is trained per hydrological regime. In these two models, for the Mediterranean and Uniform regimes lookbacks of 60 and 730 [day] are respectively taken into

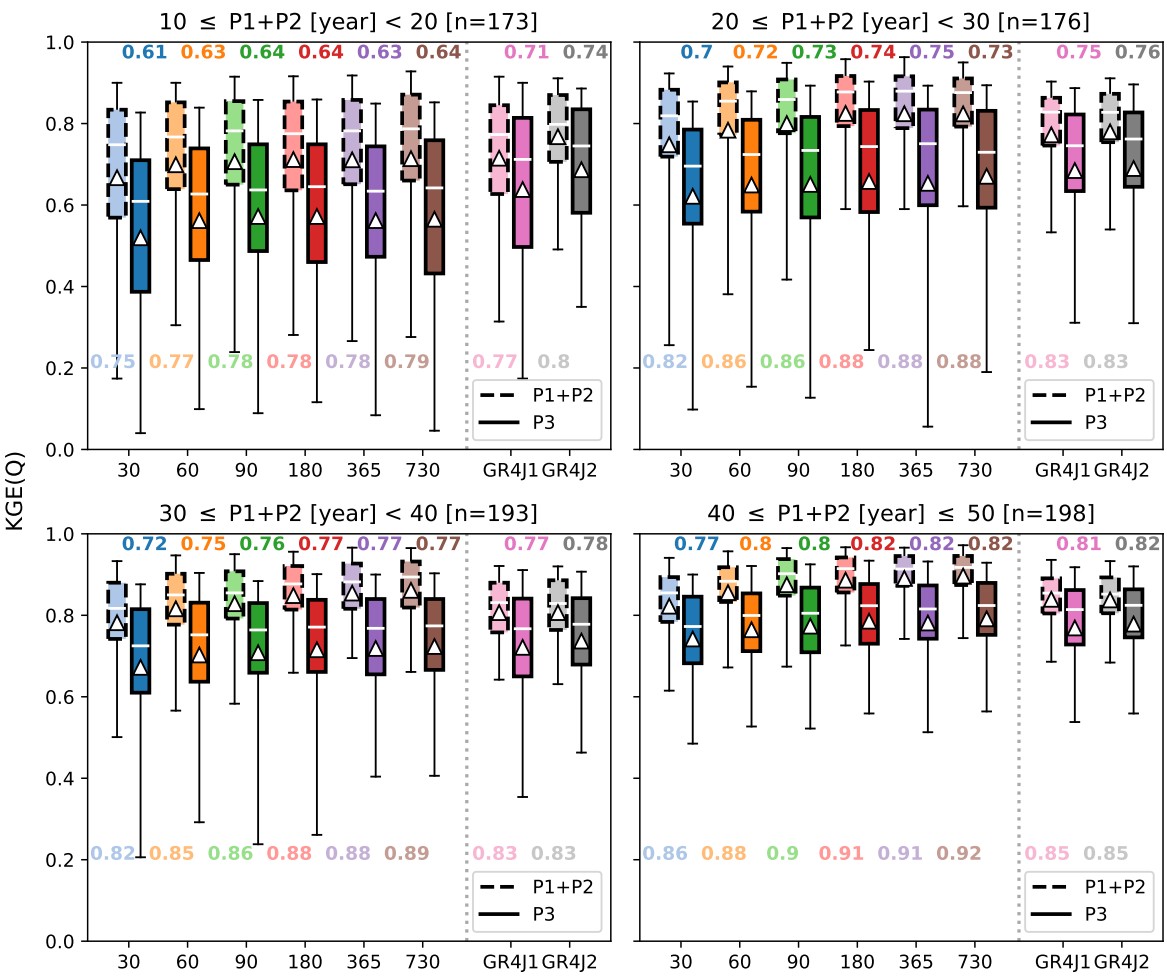

**Figure 8.** Variation of KGE scores for the catchments in the FR sample for different lookbacks of the local LSTM compared against the benchmark GR4J model. For each model, performances corresponding to the period P1+P2 (box plots with dashed edges and brighter colors) are paired up with performances corresponding to the P3 period (box plots with solid edges and darker colors). In each pair, the median KGE of the P1+P2 and P3 periods is indicated below and above the corresponding box plot, respectively. Numbers on the x-axis represent lookbacks. The results obtained by the LSTM are separated by a vertical dotted gray line from the results obtained from the GR4J model. The GR4J1 and GR4J2 models refer, respectively, to GR4J calibrated on P1 and P1+P2. The (△) symbol marks the mean KGE.





**Figure 9.** Variation of KGE scores within the A sample and across its regimes. In each regime, performances of the local LSTM model for different lookbacks (numbers on the x-axis) are benchmarked against the GR4J model calibrated on P1 (GR4J1) and P1+P2 (GR4J2). Box plots with dashed edges and brighter colors correspond to the P1+P2 period and box plots with solid edges and darker colors correspond to the P3 period. The median KGE of the P1+P2 and P3 periods is indicated below and above the corresponding box plot, respectively. The (△) symbol marks the mean KGE.





account and for all other regimes a lookback of 180 is used. Figure 10 compares the results obtained from the different regional models taking into account all catchments in the B1 sample (the upper left panel) as well as catchments within each
hydrological regime (other panels). Looking at the results in the upper left panel where all catchments (in the B1 sample) are taken into account, the two following main observations can be made. The first is based on training approach and the second is based on configuration of static features.

1. When the configuration of static features is kept the same, which is the case for the (R1, R3) and (R2, R4) pairs, the per-regime training approach outperforms the all-together training.

2. For models with an identical training approach, which is the case for the (R1, R2) and (R3, R4) pairs, the incorporation of climatic attributes ($IQ$, $IP$, and $T_{\min}$) improves performances.

Investigating the results within different regimes, almost the same pattern is found in the Uniform, Oceanic, Mediterranean, and Nival-Pluvial regimes. This pattern of performance improvement from the R1 model to the R4 model, which is due to both per-regime training and inclusion of climatic attributes, is in particular noticeable within the Uniform regime. This is while, in
the Nival regime, neither of these two factors brings performance improvement on unseen data (P3) when compared against performances of the R1 model.

Overall, the results suggest that training hydrologically similar catchments together and incorporation of climatic attributes $IQ$, $IP$, and $T_{\min}$ in static features make regional training of the LSTM more effective. This finding is interesting at least for two reasons. First, when the train catchments consist only of those with identical regimes, data from fewer catchments is
obviously used in model training and this may seem to impair learning. However, the model in turn does not have to learn divergent hydrological behaviors. Therefore, overall, learning is enhanced. Second, although the three hydroclimatic indices $IP$, $IQ$, and $T_{\min}$ have been already used in classification of catchments, that does not replace the information they bring as static attributes. We think that this is because they account also for inter regime differences (between catchments of the same regime).

## 5 Discussion

### 5.1 Optimal lookback

The results from this research suggest a relation between the optimal lookback of the LSTM and catchment memory. In the Oceanic, Nival, and Nival-Pluvial regimes, a lookback of 180 [day] reproduces adequately the hydrological response of catchment and the LSTM does not need to work through input sequences as long as 365 [day]. Two particular cases of the
optimal lookback are however identified in the Mediterranean and Uniform regimes featuring high frequency and low frequency dynamics, respectively.

In Mediterranean catchments where a lookback as short as 60 [day] found to be optimal, internal states (e.g. soil moisture) do not depend on long antecedent periods compared to catchments in other regimes. This could be explained by the (flash) flood





**Figure 10.** KGE scores for the regional R1, R2, R3, and R4 models. The results in the top left panel correspond to all catchments in the sample (B). In the other panels, these results are separately presented for each hydrological regime. In each panel, the results obtained from the local LSTM with the similar lookback are also shown (the leftmost pair of box plots). The two rightmost pairs of box plots correspond to the local GR4J models trained on the P1 (GR4J1) and P1+P2 (GR4J2) periods, respectively.





generating nature of precipitations in this regime which are particularly intense in the autumn (Fig. 5). Those precipitations can
be so intense that rapidly saturate soil after the dry summer season or getting to the point that exceed the infiltration capacity
of soil without even saturating it. Conversely, far longer lookbacks (up to 730 [day]) turn out to be required for an adequate
simulation of runoff in catchments within the Uniform regime. These catchments operate under long term dynamics and their
response can not be clearly correlated with precipitations in the present (Fig. 5). Uniform catchments occur mainly in areas
known to be highly influenced by large aquifers, such as aquifers of the Seine or the Somme River basins in the north of France
(Fig. 4). Such aquifers can significantly modify the temporal dynamics of the impacted catchments. Runoff at outlet of Uniform
catchments can therefore depend on precipitations of a long time ago, up to 730 [day].

## 5.2   Local training versus regional training

This research compares different training approaches, from the one using only individual catchments (local training) to the one
using the catchments within the same regime (regional training in the R3 and R4 models) to the one using all catchments of
a sample from different regimes (regional training in the R1 and R2 models). The effectiveness of regional training turns out
to depend on both the hydrological similarity of train catchments and the type of the incorporated static features. Therefore,
not all regional trainings outperform necessarily the local training. For instance, comparing the median KGE score in Fig. 10,
the local LSTM outperforms the regional R1 and R2 models in almost all regimes except Nival where the R2 model shows
slightly better performances. This may appear to be counter-intuitive taking into account the results from the previous studies
by Kratzert et al. (2018), Gauch et al. (2021), and Lees et al. (2021) where regional training outperformed almost always local
training. A possible explanation would be the choices made for local training in these studies. In Kratzert et al. (2018) the local
LSTM is trained using 15 years of data and no data is used for validation. Furthermore, the number of training epochs is set to
a unique value (50) for all catchments. In Lees et al. (2021), although a validation period is taken into account, it is limited to
5 years and the length of train data does not go beyond 11 years. This is while we use from 10 to 40 years of train data in local
training to obtain the results shown in Fig. 10. Moreover, we use 10 years of full record data to validate learnable parameters of
training. Besides, the required number of training epochs is not a predefined value and the early stopping criterion controls it.
Taken all together, some constrained choices are made in previous studies due to the size of the CAMELS and CAMELS GB
datasets as well as the simulation framework of the studied benchmark models. These choices are susceptible to disadvantage
the local LSTM in these studies.
In regional training, if catchments belong all to the same hydrological regime and if the climatic static features are incorporated
(the R4 model) overall performances slightly improve when compared with the local LSTM (the upper left panel in Fig. 10).
Making this comparison within the regimes, the R4 model outperforms in the Oceanic and, in particular, Uniform classes. The
local LSTM and R4 models perform equally well on unseen data of the P3 period in the Nival and Nival-Pluvial regimes. For
Mediterranean catchments, however, it is the local LSTM that outperforms. Additional static attributes may still be needed in
these last 3 classes to account for the underlying inter regime features.





## 5.3 The LSTM versus GR4J

The overall performance of the R4 model (the best performing LSTM) and the GR4J model is almost equivalent (the upper left panel in Fig. 10). The R4 model outperforms the GR4J model on the P1+P2 period but the performance distribution of GR4J is less dispersed on unseen data (P3). Comparing the median KGE score on the P3 period within each regime (the remaining panels in Fig. 10), the R4 model gives slightly higher values in the Oceanic, Nival-Pluvial, and Nival regimes. In the Nival regime, the median KGE score for the R4 model is more distinctly higher. In the Uniform regime, both models have the same median KGE score. Regional models in previous studies (Kratzert et al., 2019b; Lees et al., 2021) achieved better overall performances when compared against conceptual models. One possible reason is that, in addition to its robustness, GR4J is explicitly designed to deal with water losses and gains through an exchange parameter (Perrin et al., 2003). This is not the case for the R-R conceptual benchmark models in previous studies (Sacramento Soil Moisture Accounting (SAC-SMA), FUSE, mHM, ARNOVIC, TOPMODEL and PRMS). For instance, Lees et al. (2021) discuss that when the water balance closure is not satisfied, the LSTM performs better than their tested R-R models. Figure 11 shows a diagnostic plot of runoff coefficient ($= \frac{Q}{P_{\text{tot}}}$) versus the wetness index $WI$ ($= \frac{P_{\text{tot}}}{PET}$) for every one of the 361 catchments in sample B1, where $PET$ is obtained using Oudin et al. (2005)'s formula. The points (catchments) are colored by the KGE score. Between the 361 catchments plotted in each panel of Fig. 11, 9 catchments occur in zone $z_1$ (above the horizontal water limit line). Given that in this zone $Q > P_{\text{tot}}$, there is a surplus in its water balance and it does not therefore close. The $z_2$ zone (between the horizontal and curved lines) contains 255 catchments in which the water balance is satisfied. Finally, 97 catchments are found in the $z_3$ zone (on the right side of the curved line) where the water balance does not close since $\frac{Q}{P_{\text{tot}}} < 1 - \frac{1}{WI}$ and therefore $Q < P_{\text{tot}} - PET$ indicating a potential water deficit. The mini plot inside each panel of Fig. 11 shows the KGE score of the catchments located in each of the $z_1$, $z_2$, and $z_3$ zones as well as their median value. In the two $z_1$ and $z_3$ zones, where the water balance is not satisfied, the GR4J model has almost the same median score as that of the $z_2$ zone in which the water balance closes. In contrast, the median score for both local and regional LSTM is lower in $z_3$ than in $z_2$ but the R4 model performs better than the local LSTM. For catchments occurring in the $z_1$ zone, the median score is improved for the local LSTM but not for the R4 model. In summary, comparing performances of each model within different zones and comparing performances between the models, the following observations can be made:

1. Performances of the GR4J model in zones with water imbalance (gains and losses) are practically not different from performances in the zone with water balance closure and the local and regional LSTM do not show such robustness. In particular, in zone $z_3$, the GR4J model outperforms the two LSTM models. This behavior is in contrast with the corresponding finding of the previous study by Lees et al. (2021).

2. Taking into account all 106 catchments having either a water surplus (9 catchments) or deficit (97 catchments) in their water balance, the GR4J model outperforms the two LSTM models: 0.79 versus 0.76 (the local LSTM) and 0.78 (the regional LSTM). This does not agree with the observed overall outperformance of the LSTM over the four conceptual models in Lees et al. (2021).

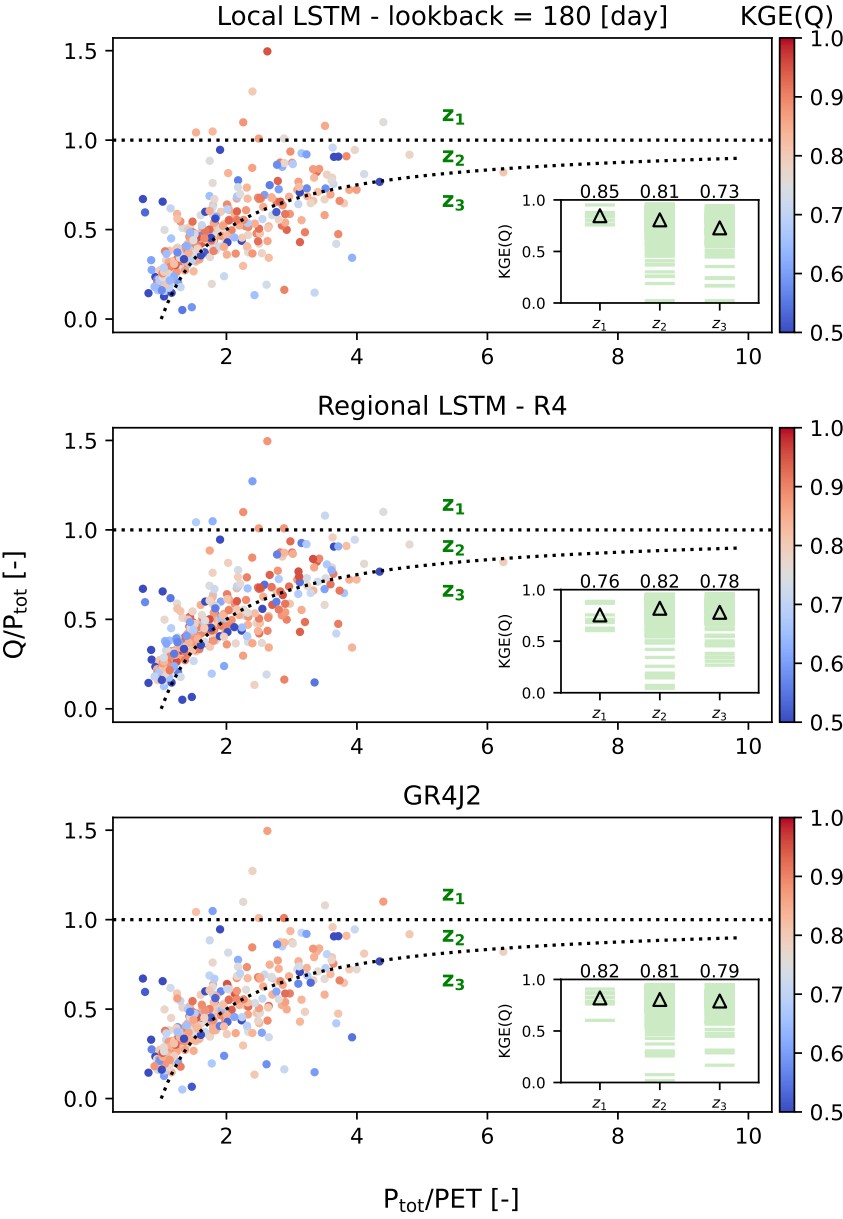

**Figure 11.** Runoff ratio ($\frac{Q}{P_{\text{tot}}}$) versus the wetness index ($\frac{P_{\text{tot}}}{PET}$) for the 361 catchments in sample B1. Scores lower than 0.5 are shown in the same color as the lower boundary of the color bar. The mini plot inside each panel of Fig. 11 shows the KGE score of the catchments located in each of the $z_1$ (above the horizontal water limit line), $z_2$ (between the horizontal and curved lines), and $z_3$ (on the right side of the curved line) zones. The ($\triangle$) symbol and numbers in the mini plot represent the median KGE of the 3 zones.





3. The decline of performances of the regional LSTM in catchments with water imbalance is consistent with the corresponding finding in Lees et al. (2021).

To further investigate about favorable performing conditions for the local LSTM (lookback=180), regional LSTM (the R4 model), and the GR4J models, Spearman's rank correlations between different catchment attributes and KGE scores of these models are plotted in Fig. 12 following the presentation of Lees et al. (2021). Between catchment slope ($S$), drainage density ($DD$), median altitude ($Z_{50}$), and surface area ($A$), the last is the only attribute that shows significant correlations with KGE scores for the local and regional LSTM models as well as the GR4J model. These correlations are positive for all models and higher for the LSTM-based models. One possible explanation for this is offered by the combined effect of two contributing factors:

1. The response time of catchment increases with increasing its surface area and the models appear to be better able to predict slow responses of larger catchments than fast responses of smaller ones.

2. With increasing catchment size, potential uncertainties associated with data of the SAFRAN product decreases since uncertainties due, for instance, to extreme events at a too small ($\ll$ 8 [km] $\times$ 8 [km]) and/or a too localized scale, would be distributed over a larger area.

Performances of the GR4J model are uncorrelated with all other attributes indicating that improvement of GR4J performances is independent of different conditions of those attributes.

In contrast, there are hydrological and climatic catchment attributes that turn out to have different degrees of correlation with KGE scores of the LSTM in a positive/negative direction. Between static features $IP$, $IQ$, and $T_{\min}$, the negative correlation of $IP$, which is stronger for the regional LSTM, confirms previous observations made on the LSTM's poorer performances in the Mediterranean regime, which were worse for the R4 model (Fig. 10 panel Mediterranean). $IQ$ shows a positive correlation with KGE scores of the local LSTM. This indicates that when $IQ$ tends to increase, performances tend to improve, which is consistent with the better performances in regimes with higher $IQ$s compared against the Uniform regime having $IQ$s lower than 0.5 (Fig. 10 panel Uniform). Taking into account the remaining hydro-climatic attributes, the wetness index ($\frac{P_{\text{tot}}}{PET}$) and $P_{\text{tot}}$ show both positive correlations for the local and regional LSTMs, indicating better performances of the LSTM in wetter climates. $Q$ shows a correlation for the local LSTM and it is expected that $\frac{Q}{P_{\text{tot}}}$ does so given the positive correlation of $P_{\text{tot}}$, which is the case.

## 5.4 Influenced catchments

In the presence of dams, the LSTM clearly outperforms the GR4J model when the degree of influences is important. Figure 13 compares performances of the (local) LSTM and GR4J in two cases: when catchments are not highly influenced (sample B1, the left panel) against performances when high $d_i$ values occur (sample B2, the right panel). Comparing the median KGE score, the local LSTM and the GR4J model perform equally well in catchments that are not highly influenced. However, for highly influenced catchments ($0.1 \leq d_i \leq 8.56$) the local LSTM outperforms clearly. We think that the LSTM detects dam behavior



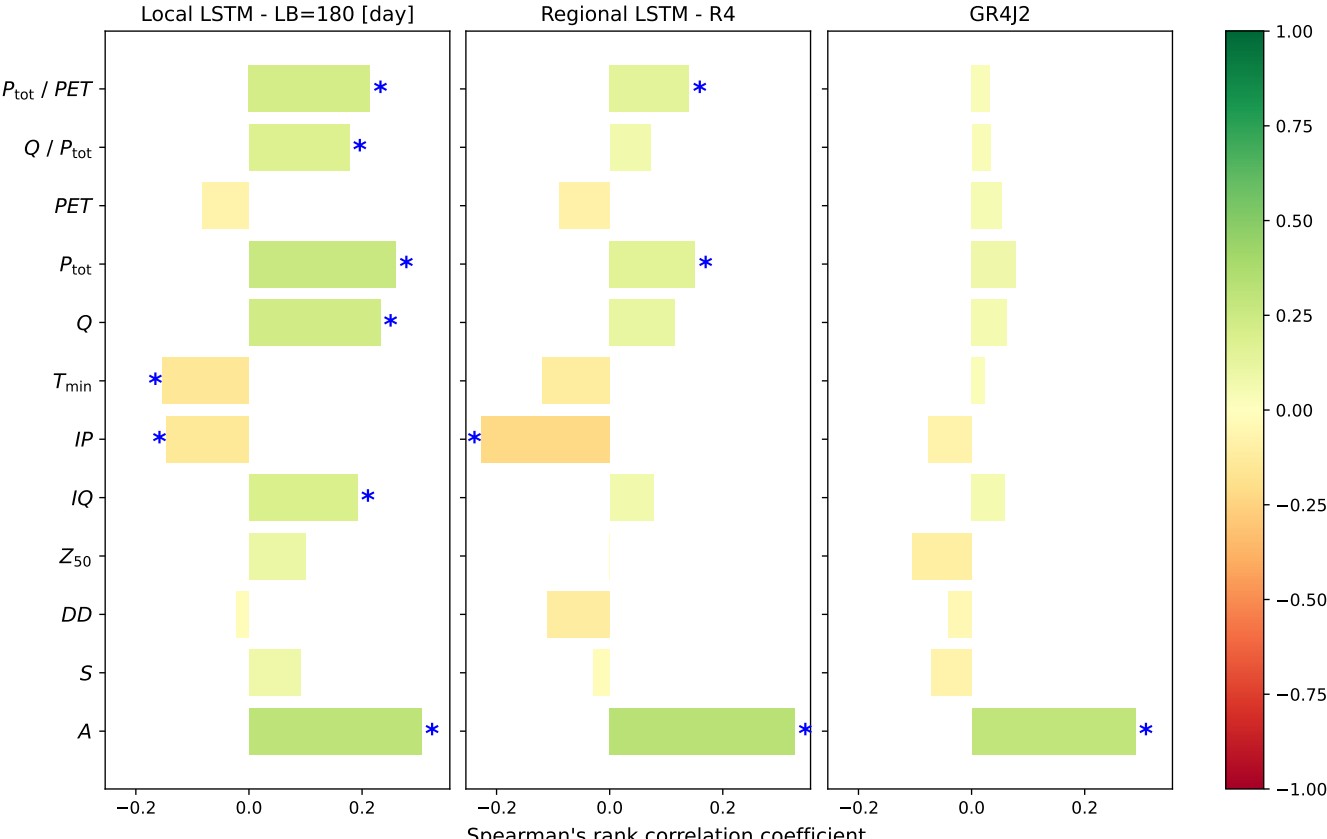

**Figure 12.** Spearman's rank correlation coefficient of different catchment attributes with KGE scores (of period P3) for the local LSTM (lookback=180), the regional LSTM (R4), and the GR4J model. The significance threshold is set to 0.01 (Lees et al., 2021). Correlations having a p-value lower than this threshold are considered to be significant and are marked by (∗). The non marked correlations are neglected. $A$: surface area [km$^2$], $S$: median slope [%], $DD$: median drainage density [%], $Z_{50}$: median altitude [m], $IQ$ [−], $IP$ [−], $T_{\min}$ [°C]: regime classification indices, $Q$: mean annual discharge [mm per year], $P_{\mathrm{tot}}$: mean annual total precipitation [mm per year], $PET$: mean annual potential evapotranspiration [mm per year].

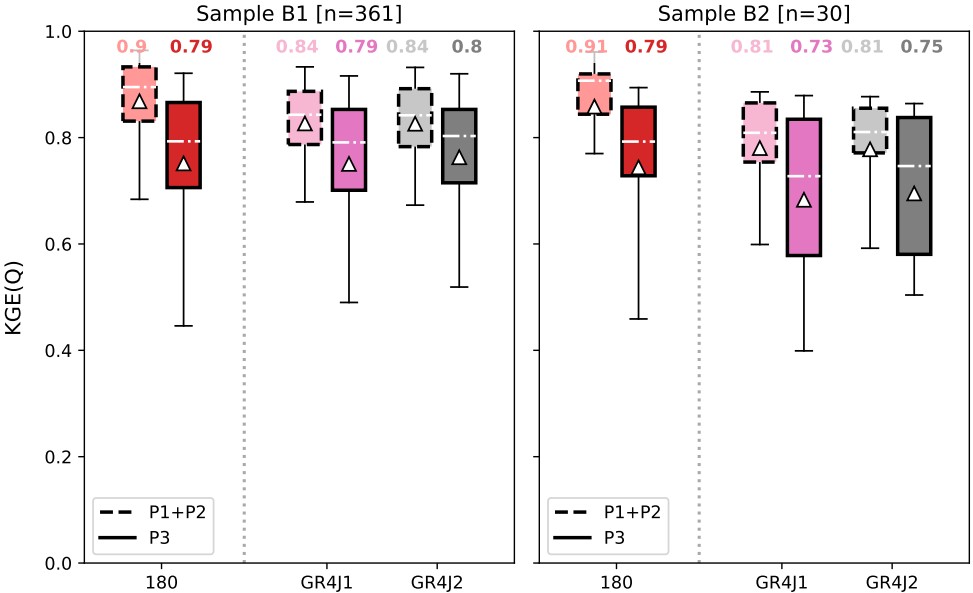

**Figure 13.** Variation of KGE scores with respect to degree of influences for the local LSTM (lookback=180) and the GR4J model. In the left and right panels, results corresponding to catchments with $d_i < 0.1$ (sample B1) and $d_i \geq 0.1$ (sample B2) are shown, respectively. The number on the x-axis indicates lookback of the LSTM model. GR4J1 and GR4J2 refer, respectively, to GR4J calibrated on P1 and P1+P2. Box plots with dashed edges and brighter colors correspond to the P1+P2 period and box plots with solid edges and darker colors correspond to the P3 period. The median KGE of the P1+P2 and P3 periods is indicated below and above the corresponding box plot, respectively. The ($\triangle$) symbol represents the mean value.

of highly influenced catchments as a change in periodicity of discharge signal and learns it from discharge time series. That is why it is able to do so independently of dam characteristics and purpose, which are not intentionally taken into account in this research to demonstrate this ability of the LSTM.

## 6    Conclusions

In this study, we used a large sample of 740 gauged catchments in the diverse hydrological context of France. Our goal was to exploit catchment hydrological features, represented by catchment regime, in training LSTM-based runoff models. We investigated two training approaches: 1) local training, i.e. when the LSTM is trained on individual catchments, 2) regional training, i.e. when the LSTM is trained on a group of catchments. In local training, we aimed to relate catchment memory to the optimal lookback of the LSTM, i.e. the size of the sequence of past forcing data that the LSTM needs to work through

to adequately reproduce runoff. In doing this, we assumed that catchments within the same regime had the same memory, therefore, catchment memory would vary only between regimes. Our results showed that so did the optimal lookback and it was regime dependent. In regional training, we investigated how hydrological homogeneity of train catchments (compared to




their number) contributed to the LSTM's learning ability. We also studied regional performances as a function of the type of static attributes used in regional LSTM models. For this purpose, we examined two classes of them: hydro-climatic (such as minimum monthly temperature) and physical (such as surface area). The results of this study suggest the following conclusions:

1. The optimal lookback varies based on the regime of catchment. It more specifically depends on the term of the hydrological processes by which regimes are featured. We identified a lower (60 [day]) and an upper (730 [day]) boundary for the optimal lookback. These boundaries occur in the Mediterranean regime (featured by intense rainfall events) and Uniform regime (featured by underground water effects). The results also showed that regimes featured by the processes longer than the processes of the Mediterranean regime but shorter than the processes of the Uniform regime have an identical optimal lookback lying between the two boundaries ($60 \leq 180 \ [\text{day}] \leq 730$).

2. When comparing performances of a regional LSTM with a local LSTM, one thing should be taken into account: the size of train (and validation) set(s) based on which the local LSTM is trained. If the local LSTM is trained (and validated) on abundant data it can have performances as good as a regional LSTM model that is trained on many catchments.

3. When training regional LSTM models, two factors make major contribution. First, the size of the train data, therefore, the number of train catchments. Second, and more importantly, the hydrological homogeneity of them. Training a regional LSTM on more catchments but with divergent hydrological regimes makes training less effective.

4. When training a regional LSTM on catchments belonging to the same regime, there is still one thing that turns out to play a role: the choice of static attributes. We think that the way they contribute is by making up for the information that is lost when passing from local to regional scale that can account for inter regime differences between hydrologically similar catchments.

5. In catchments influenced by dams, when the storage volume of reservoir per surface area of catchment ($d_i$) is important, the LSTM is better able to reproduce runoff than the conceptual benchmark model. This suggests that the LSTM is able to learn the behavior of these catchments from only discharge time series and it does so independently of dam characteristics and purpose.

The current conclusions are drawn in the French climatic context. Validating these conclusions in a different climatic context (e.g. using the available CAMELS data for the US, Great Britain, or Brazil) would help to suggest more widely applicable training approaches. Another future research direction would be to investigate systematically how performances of the two training approaches change as the train data is progressively reduced in size. The limit case of this would be to apply regional models to catchments that are not used in training and compare their performances with performances of regionalized conceptual models (Oudin et al., 2008).





**Figure A1.** KGE scores of period P3 in EXP0 for the S1 architecture involving a single standard LSTM layer of 64 nodes. The 15 grids correspond to the 15 catchments of the experiment EX0 and are arranged in five rows according to the five hydrological regimes. Regimes are indicated on the left side of the rows. In each grid, batch size and lookback increase from left to right and from bottom to top, respectively.





**Figure A2.** KGE scores of period P3 in EXP0 for the S2 architecture involving two standard LSTM layers of 32 nodes. The 15 grids correspond to the 15 catchments of the HP sample and are arranged in five rows according to the five regimes. Regimes are indicated on the left side of the rows. In each grid, batch size and lookback increase from left to right and from bottom to top, respectively.



## Appendix A: Results of hyperparameter tuning

*Author contributions.* RH and PJ designed all experiments. RH conducted all experiments and analyzed results with advice from PB, PAG, and PJ. RH and PJ prepared the first draft of the article. PB proposed the idea of a regime based analysis and gave guidance on data and

GR4J simulations. PAG proposed the study sample. PJ supervised the work and was in charge of the overall direction. All authors discussed the results and contributed to the final version of the manuscript.

*Competing interests.* The authors declare that they have no conflict of interest.

*Acknowledgements.* The authors would like to thank Météo France (https://www.data.gouv.fr/en/organizations/meteo-france/) and the Banque HYDRO (http://hydro.eaufrance.fr/) for providing the SAFRAN forcing data and runoff records used in this study. The authors would also

like to express thanks to the HYCAR research group at INRAE for extraction of the forcing and hydrometric data from their original source and the related data processes, as well as, to Jéremy Verrier and Rémi Cresson for their help with provision of CPU and GPU resources for conducting the experiments of this work.



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
