# Peer review of "How can we benefit from regime information to make more effective use of LSTM runoff models?"

_Hydrology and Earth System Sciences, 2021_

## Referee Comment (RC1)

**Review hess-2021-511**

**TITLE**

How can regime characteristics of catchments help in training of local and regional LSTM-based runoff models?

**RECOMMENDATION**

Accept after minor corrections

**REVIEWER**

John Quilty

**GENERAL COMMENTS**

This paper carefully studies long short-term memory networks (LSTM) for rainfall-runoff prediction, using a large-sample of catchments in France. The key focus is on exploring local and regional models as well as the impact of the 'lookback' period, an important hyper-parameter of LSTM, with respect to predictive performance and physical understanding of the model results. The authors include well-thought out experiments to identify the impact of the lookback period and cases where local and regional LSTM models are best suited. The authors also benchmark LSTM with GR4J, due to its useful ability to capture ground water exchanges with aquifers and/or between catchments.

The authors spend a considerable amount of effort on tying the performance of LSTM, locally and regionally, to a physical understanding of the results. Some examples include the comparison between local and regional LSTM models with GR4J in terms of a water balance exercise in Section 5.3 as well as the ability of LSTM to predict runoff in controlled catchments at a higher degree of accuracy than GR4J (in Section 5.4). This paper also presents findings (e.g., LSTM does not necessarily outperform simple conceptual rainfall-runoff models) that are counter to other recent studies on LSTM (Gauch et al., 2021; Kratzert et al., 2019; Lees et al., 2021); in all such cases, the authors take the time to carefully describe potential reasons for these differences.

Overall, this paper is very strong and I could not find much to criticize. The methodology seems correct. The figures are very nice and easy to interpret and I did not find any of the content, tables, or figures to be superfluous.

I suspect this paper will be very useful to other researchers interested in exploiting the generality of machine learning for hydrological modelling and rainfall-runoff prediction, in particular. I think the paper only needs some very minor corrections and some additional brief explanations (as noted below). Afterwards, the paper could be published.

**SPECIFIC COMMENTS**

1. Line 32: does LSTM also help mitigate against exploding gradients? If so, this would be good to mention as well.

2.  Fig 1: it would be good to include a description of the acronyms HP and FR in the figure caption (since it is unclear what these acronyms represent).

3.  Grammatical corrections: for the most part, the paper is well-written but there are a number of grammatical errors. I stopped correcting such errors around line 154. I recommend that a very carefully read through the paper be completed before re-submission.

4.  The sentence on line 165 can be moved to the last sentence of the same sub-section. A short sentence, 'The main equations used in LSTM are as follows (Ref, XXX):' can be used in it's place.

5.  L205-206: is this sort of standardization the most appropriate for LSTM? Since sigmoid and tanh activation functions are used, should not the data be scaled to [0,1] or [-1,1] as these ranges match the output ranges of the (previously mentioned) activation functions? Perhaps others have adopted the form of standardization adopted here, if so, can the authors indicate this?

6.  What were the hyper-parameters (alpha, beta_1, beta_2) set to in the Adam algorithm?

7.  Equation 17: what does epsilon represent?

**TECHNICAL CORRECTIONS**

- Line 16: '…**a** catchment…'
- L35: 'significantly depends'
- L57: 'It is therefore tried to…' Unclear, please re-write this sentence.
- L61: 'should' instead of 'can'.
- L73: 'popular' instead of 'acknowledged'?
- L135: 'The pattern of runoff…'
- L148: 'that have' instead of 'and has'.
- L149: '…**are** unknown lags between the response of the system **and** a continuous input to it.'
- L150: 'signal**s**'
- L154: 'In an ordinary RNN**,** cell information sharing is **achieved** through a feedback connection…'
- Fig 6 caption: '…are computed **by** taking…'

**REFERENCES**

Gauch, M., Mai, J., Lin, J., 2021. The proper care and feeding of CAMELS: How limited training data affects streamflow prediction. Environ. Model. Softw. 135, 104926. https://doi.org/10.1016/j.envsoft.2020.104926

Kratzert, F., Klotz, D., Shalev, G., Klambauer, G., Hochreiter, S., Nearing, G., 2019. Towards learning universal, regional, and local hydrological behaviors via machine learning applied to large-sample datasets. Hydrol. Earth Syst. Sci. 23, 5089–5110. https://doi.org/10.5194/hess-23-5089-2019

Lees, T., Buechel, M., Anderson, B., Slater, L., Reece, S., Coxon, G., Dadson, S.J., 2021. Benchmarking data-driven rainfall--runoff models in Great Britain: a comparison of long short-term memory

(LSTM)-based models with four lumped conceptual models. Hydrol. Earth Syst. Sci. 25, 5517–5534. https://doi.org/10.5194/hess-25-5517-2021

---

## Referee Comment (RC3)

I would like to respond briefly to the author's responses before this goes back to the editor. I think this is important to state up front that these responses are not sufficient to warrant publication of this paper (in my opinion). The authors' responses indicate a very deep misunderstanding of the models that they are working with, and also a deep misunderstanding of relatively simple and basic ML procedures. I sincerely hope that they will take the advice I gave in the original review, because this paper is not close to being publishable as it stands, and the authors responses only further indicate that we are not approaching a reasonable experiment.

(author comments in purple)

*As the reviewer confirms in their next comment, the only benefits we got from the hyperparameter tuning experiments were deciding to use what model structure with what batch size.*

This is because you did not do hyperparameter tuning correctly. You did not tune most of the important parameters, and you did not do a full sweep. Indeed, hypertuning an LSTM is extremely important and will change the results significantly. It is clear from what you reported that you did not do this correctly, however the fact that you did not find it useful is by itself an indication that you did not do it correctly. Hypertuning is an extremely basic part of any ML workflow, and the fact that such a basic thing was done so poorly is a hint that there may be other serious issues with this study (indeed there are many).

Hypertuning must be done with a full sweep of parameters on a large dataset separately for each model. This is non-negotiable. If this is not done then the results of the study are meaningless. Hypertuning matters – it is a problem that the authors are seeing/arguing otherwise.

*We would assume that the reviewer's term "LSTM's behavior" could be translated to "LSTM's performance measured by the median KGE"*

No, this is not correct. What I mean is about how the activation functions in each gate "learn" to span the range of possible inputs and responses. I suspect from reading this response that the authors have not spent much, if any, time understanding the behavior of the gate structures in their model. Have the authors plotted or visualized their activation functions? Or their forget, input, and output gates? Have they spent any time looking at the range and independence of their cell and hidden states? I suspect not. Please understand that several years worth of this kind of effort is underneath all of the studies published by Kratzert et al.

*Our reading of this is that in order to "maximize" the benefits of hyperparameter tuning over the entire population, more than 15 catchments should be taken.*

No, again this is not a correct interpretation of my comment, and it (again) belies a deep lack of understanding about basic ML workflows. You should hypertune on a sample that is as

representative as possible of the use case. In the case of an LSTM for streamflow, you will use large, diverse datasets to train the model, so hyperparameter tuning will be done on similarly large, diverse datasets. For local models, you will hypertune each individual local model separately on held-out data from that catchment.

*Above all, please note that the way that hyperparameter tuning is performed in this paper does not allow — on its own — to be carried out for all catchments.*

Yes, this is because you've done hyperparameter tuning incorrectly. This is a flaw in your workflow that needs to be fixed before the results can be meaningful or trusted. Please see other studies (preferably by ML experts, not by other hydrologists) on how to do correct hypertuning with the LSTM for streamflow. Kratzert uses three data periods for each catchment. There are alternative ways to do this, but the hyperparameter tuning must be done in a meaningful way, and it must be done separately for each model that is benchmarked/compared. None of that is done correctly in this study.

About cell state size, you said: *"It would be therefore arguable that the most effective hidden unit number would be always much higher than 64 since Lees et al. (2021) reported that this rule did not hold true in their work."*

First, why are you trusting a secondary source over a primary source? Kratzert and Hochreiter are ML experts and Lees is not. Kratzert and Hochreiter spent years developing, training, and tuning the model, and Lees did not. Lees did a good job, but his paper is just an application study. Kratzert has spent years building and running operational versions of this model at local, regional, and global scales for many different purposes (operational models, research models, private sector, government, etc.), as well as exploring the model in depth in a series of publications. Kratzert worked directly with the world-leading LSTM expert. Lees' paper is a one-off application study. I'm not saying that Lees' study is wrong or flawed (it is a good study), but I wonder why you would choose that one in particular to model your experimental design from. In response to your comment about publication date, Kratzert has many papers newer or contemporary with Lees. Leaving aside the question of how you choose which references to model your study after, in the present case you have no idea what cell state size to use because you didn't test this parameter. No matter how you slice it, it is inappropriate to not hypertune one of the most important parameters in the model.

*"Compared to local models trained on individual catchments, regional models are expensive to develop and maintain — the slightest change in their setup necessitates re-training a huge model, even if we do not start from scratch."*

This is not true. This sounds like things we very often hear from hydrologists who do not have very much or any experience using these deep learning models. There is very little overhead for using a regional model that does not also exist for using local models, and an extremely large amount of overhead for using local models (fine tuning is cheaper than hypertuning). Training regional models is not expensive, and can be done on a laptop (unless you design your code

inefficiently, however you should be using the NeuralHydrology github repository unless you are an expert at building ML models – there is a huge amount of work underneath setting one of these models up correctly, and it is easy to do it quick and wrong).

I response to this comment, if/when the authors actually become familiar with these DL models and start using them regularly, you will find three things:
(1) There is no significant difference in the "size" of regional vs. local models (the difference in size is a few MB, and the increase in the number of free parameters does not cause performance or behavior issues).
(2) It is not harder to use regional models than local ones (in fact, the opposite because fine tuning is cheaper than hypertuning).
(3) Regional models are usually more accurate (when done correctly) because any regional model can be tuned as necessary.

*"The issue of computational cost of regional training is tied to another question: availability of national databases, which is often not the case — at least to public."*

This is irrelevant, because there is enough public streamflow data to train a base model. There is no reason to use "national" data because you can just tune your base model. Use the largest dataset possible. This, again, is an imagined constraint that we hear often from hydrologists who do not have a strong understanding of how these models work. I'm going to be a little personal here and say that it is frustrating to respond to these types of comments, when it is extremely clear that the authors don't actually know this from experience (I'm so sorry to be direct, but when the authors gain experience, they will see that this statement is incorrect).

Train your base model on a large-sample datasets, and then do whatever you want (i.e., tuning, etc.) with those base models using whatever dataset you are actually interested in. This is how these models work best. We have a lot of experience doing this (including operational modeling at all scales from single-catchment to global).

*"There are real world cases of catchment for which it would not be appropriate to use universal models. Dam influenced catchments are one of these cases"*

This is, again, not really correct. I want to emphasize that there is no well-established way to deal with dams and reservoirs in (any type of) hydrology model, including deep learning (this is a problem that we are currently working on). The problem is that the authors did not make this comment from any actual experience using LSTMs. I know this not only because their use of LSTMs throughout this paper, and as indicated in their replies, is in many places contrary to (well-known) best practices, however, I also know that the authors are not making this claim from experience because I've spent months working on this (modeling dams with deep learning), and locally trained models are one of the worst options that we have tried.

Again, I'm so sorry to be personal about this, but I am really frustrated at having to respond to comments made by hydrologists with no actual, meaningful experience with these issues. I am

asking (begging) hydrologists to stop imagining things, and make their decisions about which and how to use different models based on evidence and experience. I'm so sorry to be direct, but I'm asking the editor and authors to please understand how pervasive and pernicious this problem is in the deep learning hydrology community (pretending to understand the limitations of different models without actually exploring those limitations in depth). I would welcome a published study comparing local vs. regional models for managed catchments, but such a study will need to be done using best-practices (including rigorous hypertuning, rigorous benchmarking, and rigorous fine tuning). Such a study could (possibly) be interesting even if it *didn't* include any of the more sophisticated ways to approach the problem (e.g., reinforcement learning, transferable/tunable head layers, multi-output learning, data assimilation, etc.)

For what it's worth, we use LSTMs for managed catchments operationally at both the global and local scale in both the public and private sectors. This is after years of experimenting and practice. I'm not saying that I know the solution, but I am saying that it is transparently clear that the authors have made this claim without actually trying it in a serious way.

*"Local trainings in this paper are not carried out in the same way as any of the previous studies. Therefore, it would be arguable to immediately rule them out based on previous studies"*

Definitely, if you think you have something new please test it. I'm not seeing anything in this study that represents a new idea in terms of how to train models (we've performed and published rigorous local vs. regional LSTM studies multiple times in the past, using what appears to be an identical training procedure except that we did real hypertuning). I don't want to discourage you from testing whatever ideas you might have, but please do the experiments in a way that allows us to be sure that the effects we are seeing aren't due to taking shortcuts. If you are computationally limited (e.g., can't do full hypertuning for each model that you want to test) then this means that you can't do the experiment - full stop. Please don't publish half-done studies, it clouds the literature with misleading results. I want to be very clear: I am happy to see the authors testing these two hypotheses outlined in this paper, but they need to use an experimental design that will actually answer the questions that they want to ask.

---

## Author Comment (AC1)

**Author Response to Review of**

**How can regime characteristics of catchments help in training of local and regional LSTM-based runoff models?**

Reyhaneh Hashemi, Pierre Brigode, Pierre-André Garambois, Pierre Javelle

*HESS,* `doi:10.5194/hess-2021-511`

RC: Reviewer Comment    AR: Author Response

**Reviewer: John Quilty**

**1. GENERAL COMMENTS**

RC: **This paper carefully studies long short-term memory networks (LSTM) for rainfall-runoff prediction, using a large-sample of catchments in France. The key focus is on exploring local and regional models as well as the impact of the 'lookback' period, an important hyper-parameter of LSTM, with respect to predictive performance and physical understanding of the model results. The authors include well-thought out experiments to identify the impact of the lookback period and cases where local and regional LSTM models are best suited. The authors also benchmark LSTM with GR4J, due to its useful ability to capture ground water exchanges with aquifers and/or between catchments.**

**The authors spend a considerable amount of effort on tying the performance of LSTM, locally and regionally, to a physical understanding of the results. Some examples include the comparison between local and regional LSTM models with GR4J in terms of a water balance exercise in Section 5.3 as well as the ability of LSTM to predict runoff in controlled catchments at a higher degree of accuracy than GR4J (in Section 5.4). This paper also presents findings (e.g., LSTM does not necessarily outperform simple conceptual rainfall-runoff models) that are counter to other recent studies on LSTM (Gauch et al., 2021; Kratzert et al., 2019; Lees et al., 2021); in all such cases, the authors take the time to carefully describe potential reasons for these differences. Overall, this paper is very strong and I could not find much to criticize. The methodology seems correct. The figures are very nice and easy to interpret and I did not find any of the content, tables, or figures to be superfluous.**

**I suspect this paper will be very useful to other researchers interested in exploiting the generality of machine learning for hydrological modelling and rainfall-runoff prediction, in particular. I think the paper only needs some very minor corrections and some additional brief explanations (as noted below). Afterwards, the paper could be published.**

AR: We would like to thank you for reading our manuscript carefully and with interest and for your encouraging comments. We are very pleased to read that you find our paper useful. We would like to invite you to find our point by point response to your comments in the following sections.

**2. SPECIFIC COMMENTS**

**2.1. Line 32**

RC: **Does LSTM also help mitigate against exploding gradients? If so, this would be good to mention as well.**

AR: Thank you for this relevant question. The answer is, yes. This is because vanishing and exploding gradients both result from the same mathematical challenge when optimizing neural networks (NNs) with very high non linearities, although the latter case (exploding gradients) is less frequent Goodfellow et al. (2016)). A full description on how LSTM overcomes both vanishing and exploding gradients is given in Hochreiter and Schmidhuber (1997). To put it briefly and in very simple words, in deeply nested NNs, such as (vanilla) RNNs when lookback ($T$) becomes large, it happens that a factor — which in the problematic case is not close to an absolute value of 1 — gets multiplied by itself over and over — $T$ times — due to the chain rule of the calculus. Therefore, the result will either exponentially shrink (if the factor is initially $< 1$) or exponentially grow (if the factor is initially $> 1$) and this is where the vanishing or exploding gradient issue arises.

LSTM establishes derivatives that neither vanish nor explode. Contrary to RNNs, LSTM weights are not constant in all time steps and depend on the input of each time step ($X_t$).

Following the reviewer's suggestion, we propose to include this explanation in the new version. We also intend to rewrite this section to make it more clear and tractable. For instance, we realized that the term "sharing important information between time steps of a time sequence" that we used in Line 151 could be misleading as we did not specify what kind of parameter sharing we were referring to and the reader could confuse it with the "constant weights" in RNNs.

**2.2. Figure 1**

RC: **It would be good to include a description of the acronyms HP and FR in the figure caption (since it is unclear what these acronyms represent).**

AR: Thank you for noticing this. Following your suggestion and the point made by the Anonymous Referee #2 on our naming strategy, we plan to revise all instances of acronym/letter based names. Nevertheless, "HP" and "FR" were intended to be acronyms for Hyperparameter and France, respectively.

**2.3. Grammatical corrections**

RC: **For the most part, the paper is well-written but there are a number of grammatical errors. I stopped correcting such errors around line 154. I recommend that a very carefully read through the paper be completed before re-submission.**

AR: Thank you for spotting these grammatical errors. We will proofread the entire paper to fix the mentioned mistakes and to check for any other errors.

**2.4. Line 165**

RC: **The sentence on line 165 can be moved to the last sentence of the same sub-section. A short sentence, 'The main equations used in LSTM are as follows (Ref, XXX):' can be used in it's place.**

AR: Thank you for this suggestion. We will update this subsection as proposed.

[Figure]

(a) $\sigma(x)$ vs. $\dfrac{\mathrm{d}[\sigma(x)]}{\mathrm{d}(x)}$

(b) $\tanh(x)$ vs. $\dfrac{\mathrm{d}[\tanh(x)]}{\mathrm{d}(x)}$

Figure 1: Sensitivity of the derivative of the Sigmoid and $\tanh$ functions to the range of input data.

**2.5. Lines 205-206**

**RC:** **Is this sort of standardization the most appropriate for LSTM? Since sigmoid and tanh activation functions are used, should not the data be scaled to [0,1] or [-1,1] as these ranges match the output ranges of the (previously mentioned) activation functions? Perhaps others have adopted the form of standardization adopted here, if so, can the authors indicate this?**

AR: Thank you for this interesting question. LeCun et al. (2012) explain that why centering the input data around 0 and scaling them by the standard deviation is typically a good idea and usually makes gradient descent converge faster. Besides, we could not in principle benefit from a [0, 1] scaling since the temperature feature could include negative values.

The interesting point of standardization comes when we investigate how the derivative of different activation functions changes with respect to the range of input data. Please note that here we are talking about the activation function for the hidden layers and not the last layer, which is given by the type of the problem — Softmax for multi-class classification, Sigmoid for binary classification, and Identity for regression. Looking at Figure 1 of the present document, it turns out that the Sigmoid ($\sigma(x)$) and $\tanh$ functions suffer from a problem — their derivative gets saturated very quickly. By the term "saturation", we mean that their derivative approaches very quickly to zero indicating that weights can not get updated effectively at these points thus the NN can not learn effectively. We observe this problem almost everywhere except in the small region in the middle centered around 0 where the derivative is the most dynamic. Therefore, having the inputs centered around 0 with a variance of 1 would also help fall in the useful area of these functions. Please note that even in their dynamic region the derivatives are small and could bring about the vanishing gradient problem in NNs with high non linearities.

Now, you might ask why not simply using an activation function that does not have vanishing gradients, for instance ReLu? The answer is that the ReLu activation function proved to be typically a more appropriate default choice — if we are allowed to use it. In LSTM, there is a specific reason for which we need to stick with the sigmoid function in gates, despite the mentioned disadvantages. Indeed, the it plays a "gate" role — a function granting us a value between 0 and 1 — and it is not possible to replace it by ReLu or any other activation functions not having this output range.

Kratzert et al. (2018) mentioned that they standardized their data using the mean and standard deviation of their training data and we will indicate this information in the new version.

**2.6. Adam algorithm**

**RC:** **What were the hyper-parameters (alpha, beta_1, beta_2) set to in the Adam algorithm?**

**AR:** We kept all arguments in the Adam optimization module of the Keras library, including $\alpha$ (learning rate), $\beta_1$, and $\beta_2$ ($L^1$ and $L^2$ norms) at their default values (Chollet et al., 2015):

```
tf.keras.optimizers.Adam(
    learning_rate =0.001,
    beta_1=0.9,
    beta_2=0.999,
    epsilon=1e-07,
    amsgrad=False,
    name="Adam",
    )
```

**2.7. Equation 17**

**RC:** **what does epsilon represent?**

**AR:** Thank you for noticing this. We forgot to indicate that Kratzert et al. (2019) added this term ($\epsilon$) to the denominator in the equation of $NSE^*$ so that the loss function would not explode when $s$ was very close to 0 (catchments with very small discharge variance).

**3. TECHNICAL CORRECTIONS**

**RC:** **Lines 16, 35, 57, 61, 73, 135, 148-150, 154, Figure 6**

**AR:** Thank you for these corrections. We will update the mentioned lines and figure.

**REFERENCES**

F. Chollet et al. Keras, 2015. URL https://github.com/fchollet/keras.

M. Gauch, J. Mai, and J. Lin. The proper care and feeding of camels: How limited training data affects streamflow prediction. *Environmental Modelling Software*, 135:104926, 2021. ISSN 1364-8152. 10.5194/hess-2021-511https://doi.org/10.1016/j.envsoft.2020.104926. URL https://www.sciencedirect.com/science/article/pii/S136481522030983X.

I. Goodfellow, Y. Bengio, and A. Courville. *Deep Learning*. MIT Press, 2016.

S. Hochreiter and J. Schmidhuber. Long short-term memory. *Neural computation*, 9(8):1735–1780, 1997.

F. Kratzert, D. Klotz, C. Brenner, K. Schulz, and M. Herrnegger. Rainfall–runoff modelling using long short-term memory (lstm) networks. *Hydrology and Earth System Sciences*, 22(11):6005–6022, 2018. 10.5194/hess-2021-51110.5194/hess-22-6005-2018. URL https://hess.copernicus.org/articles/22/6005/2018/.

F. Kratzert, D. Klotz, G. Shalev, G. Klambauer, S. Hochreiter, and G. Nearing. Towards learning universal, regional, and local hydrological behaviors via machine learning applied to large-sample datasets. *Hydrology and Earth System Sciences*, 23(12):5089–5110, 2019. 10.5194/hess-2021-51110.5194/hess-23-5089-2019. URL https://hess.copernicus.org/articles/23/5089/2019/.

Y.-A. LeCun, L. Bottou, G.-B. Orr, and K.-R. Müller. Efficient backprop. In *Neural networks: Tricks of the trade*, pages 9–48. Springer, 2012.

T. Lees, M. Buechel, B. Anderson, L. Slater, S. Reece, G. Coxon, and S. J. Dadson. Benchmarking data-driven rainfall–runoff models in great britain: a comparison of long short-term memory (lstm)-based models with four lumped conceptual models. *Hydrology and Earth System Sciences*, 25(10):5517–5534, 2021. 10.5194/hess-2021-51110.5194/hess-25-5517-2021. URL https://hess.copernicus.org/articles/25/5517/2021/.

---

## Author Comment (AC2)

**Author Response to Review of**

**How can regime characteristics of catchments help in training of local and regional LSTM-based runoff models?**

Reyhaneh Hashemi, Pierre Brigode, Pierre-André Garambois, Pierre Javelle

*HESS*, `doi:10.5194/hess-2021-511`

**RC: Reviewer Comment**     AR: Author Response

**Reviewer: Anonymous Referee #2**

AR:  We would like to thank the Anonymous Referee #2 for their review and constructive thoughts, which we believe will help significantly improve our paper. The reviewer is critical of the design of our regional experiments and identified technical issues warranting a revision of these experiments. We acknowledge certain limitations and caveats regarding this aspect and agree with many of the points the reviewer makes. We plan to make the necessary revision(s), given further below, to address these issues. It seems to us that few points regarding details of our experiments are misunderstood. These points are clarified where they are elaborated on. We would like to invite the reviewer to find our point by point response to their individual comments in the following sections.

**1. SUMMARY OF REVIEW**

RC: **This paper addresses two research questions related to the use of LSTMs for rainfall-runoff modeling: (1) Does appropriate sequence length depend on hydrological regime, and (2) should LSTM training be done on hydrologically similar basins?**
**To state my opinion up front, I have run similar experiments (unpublished) and found results that are qualitatively different than what are reported here. There are several technical issues in this paper (overall, the methodology is not appropriate for testing the stated hypotheses), and it might be worth addressing those before we look carefully at the results.**

AR:  We thank the reviewer for their interest in performing similar experiments. We would be happy to engage in an ongoing dialogue with the reviewer about the details of their experiments since without further information, in particular, on the hydro-geo-climatic context of their data, it would be hard to provide a definite explanation. Indeed, such discrepancies could be investigated at different levels. At the highest level, we would conjecture that the reason lies in definition of the homogeneity component. We believe that the hydrological similarity rule — i.e., the regime classification — is a crucial question. Given the term "similar experiments", we could think of the following two cases, which both could have "very" different learned latent spaces compared with ours.

Case 1  The exact same classification is applied to a sample in a non French context. In this case, we would be afraid that the exact same rule would not be immediately applicable to other climatic contexts.

The following elaboration on our classification approach aims to underline how it is intensely context dependent — in terms of number of classes, criteria, and thresholds.

As a property of the French context, we knew in advance that there existed five main regime patterns, which we named Uniform, Mediterranean, Oceanic, Pluvial, and Pluvial-Nival. Therefore, any catchment in our sample could be classified in one of the five categories. Using the fact sheets available at https://webgr.inrae.fr/activites/base-de-donnees/, we tried to identify hydro-geo-climatic signals that could reflect different features of all five patterns. The decision feature(s) — based on which we could distinguish one pattern from the others — was (were) not the same for all regimes. For instance, we were observing that the minimum temperature attribute alone was able to detect the Nival pattern. While, in catchments with known water ground effects, it was certainly a criterion on discharge that was doing this. In the same spirit of a decision tree algorithm, but at a human level, we concluded that the mean annual discharge, total precipitation, and temperature signals would be the most useful signals to exploit to identify the distinctive attribute(s) in each class. After a number of trial and errors on different properties of these signals, such as the number, magnitude, and time of occurrence of their global/local peaks, we identified our classification attributes — $IQ$, $IP$, and $T_{\min}$ — and their thresholds. Therefore, inferring the similarity rule for any other climatic context warrants a redefinition of different elements in this analysis.

Case 2 Another classification (concluded at an AI or a human level) is applied to data belonging to a non French context. In this case, getting different results would not be surprising — since a different rule involves different decision attributes and criteria. The question in this case would be rather the extent to which we could compare the results obtained from two different classifications.

To conclude on this point, we believe that such cross study comparisons warrant caution, since the imposed similarity rule will not be identical between the studies. That is why we would like to emphasis and acknowledge that research questions established on a subjective component, such as regime classification in our paper, will always make the corresponding conclusions subject to that component. The suggestion we made in Section 6 of the paper aimed to address this point:

> The current conclusions are drawn in the French climatic context. Validating these conclusions in a different climatic context (e.g. using the available CAMELS data for the US, Great Britain, or Brazil) would help to suggest more widely applicable training approaches.

RC: **My overall recommendation is to revise the experiment as suggested in one of the comments below. The experimental design that is appropriate to test the (two) hypotheses outlined here is very simple (but somewhat computationally expensive). If the authors were to find similar results using a more appropriate experiment, this would be an interesting study.**

AR: We provide our response to this comment later in Subsection 2.3 where the reviewer details their suggested design of experiments.

**2. COMMENTS**

**2.1. Hyperparameter tuning**

RC: **Hyperparameter tuning was done on LSTMs trained on individual basins. LSTMs trained on individual basins behave fundamentally differently than LSTMs trained on multiple basins, which means that lessons learned from hypertuning on individual basins do not translate to multiple-basin models.**

AR: The reviewer identifies two unfavorable points:

Point 1 "LSTMs trained on individual basins behave fundamentally differently than LSTMs trained on multiple basins".
We would assume that the reviewer's term "LSTM's behavior" could be translated to "LSTM's performance measured by the median KGE". Taking into account each individual panel of Figure 10 [1] and comparing the median KGE of its local LSTM (the leftmost pair of box plots) and the regional LSTMs (the box plots in the center), we do not recognize any fundamental difference in their performances. Except for the Mediterranean panel, the local LSTM and at least one regional LSTM have a very similar median KGE.

Point 2 "[...] which means that lessons learned from hypertuning on individual basins do not translate to multiple-basin models".
As the reviewer confirms in their next comment, the only benefits we got from the hyperparameter tuning experiments were deciding to use what model structure with what batch size. Furthermore, we believe that we do have no evidence to confirm or reject the utility of these two tunings for regional training — we simply did not test any other variations of them (batch size and model structure) for regional LSTMs.

RC: **Additionally, 15 catchments is not enough for robust hypertuning – we would need to perform hyperparameter tuning on the full (evaluation) dataset (although see a later comment – the experimental design needs to be changed fundamentally). Also, notice that the only portions of the "hypertuning" that were actually used for the other experiments in this paper were (1) discarding the S2 model architecture, and (2) batch size.**

AR: (For the sake of convention, please let us use the terms "optimal" and "tuned" interchangeably hereafter.) We completely agree with the reviewer's point that a robust hyperparameter tuning warrants a sample larger than 15 when the population size is over 700. Our reading of this is that in order to "maximize" the benefits of hyperparameter tuning over the entire population, more than 15 catchments should be taken. This means that: to get close to the optimal point of as many of catchments as possible we need to perform tuning on more than 15 catchments. This is a sound argument. We therefore would like to clarify that our goal was not to be tuned with respect to every single point of the population. Our goal was rather to be in a broadly useful zone in the space of hyperparameters. We are aware that such zone would be close to the optimal point for some instances, far away from the tuned point for some others, and possibly at the exact same position of the optimal point for some (very rare) cases.

Above all, please note that the way that hyperparameter tuning is performed in this paper does not allow — on its own — to be carried out for all catchments. The reason is that all the results presented for our hyperparameter tuning, and therefore all the conclusions drawn form them, correspond to the evaluation period (last interval). This indicates that their evaluation data are already used in hyperparameter tuning and no unseen data are left for these catchments. That is why they need to be immediately excluded from the sample. If we did hyperparameter tuning on all catchments, we should have excluded them all — already at the hyperparameter tuning stage — and we would have had no more study catchments for our main experiments (EXP1 and EXP2). The reviewer might ask why we adopted such tuning approach. The answer is since in the main experiments models are being evaluated on the evaluation period and we believe that in hyperparameter tuning we need to do an identical evaluation.
* * *
[1] in the Preprint version

We would nevertheless like to state that we have no objection to drop the section (and any other contents) related to hyperparameter tuning, if the reviewer finds it to be superfluous.

**2.2. Number of hidden units in the LSTM layer**

RC: **There is strong relationship between the dimension of the cell state and the sequence length, and also between the cell state dimension and the ability of the model to generalize (Kratzert et al. (2019) shows how the model uses the cell state to map catchment similarity). This parameter was not included in the hyperparameter tuning, and it was also not considered in the experimental design. 64 cell states is smaller than used by most of the previously published work. The hypotheses that are tested here are about the ability of the model to generalize and about memory timescales, both of which are directly controlled by the cell state (more cell states means more ability to have different memory timescales for different hydrological regimes).**

AR: We thank the reviewer for this relevant and constructive comment regarding the number of hidden units in the LSTM layer. We also acknowledge the findings of Kratzert et al. (2019) on the relation between this parameter and model generalization ability. We believe that improving the generalization aspect, i.e. the ability of the model perform well on "unseen" data, would be of central importance/interest when applying the regional model to ungauged catchments — catchments not used in model training. Our paper did not intend to investigate any such applications. Besides, in Lees et al. (2021), they use a cell state size of 64 and the Entity-Aware-LSTM model — the model used and developed by Kratzert et al. (2019). We quote from the Preprint version of their paper (available at `https://doi.org/10.5194/hess-2021-127`):

> "We chose the hyper-parameters (dropout rate, hidden size - hs) based on the choices in previous studies (Kratzert et al., 2019)."

and in the final version (available at `https://doi.org/10.5194/hess-25-5517-2021`) they update/add that:

> "We chose the hyper-parameters (dropout rate, hidden size – hs) based on analysis of the NSE performances, finding that the improvement of further model complexity (increased hidden size) was negligible after a hidden size of 64. The hidden size was also consistent with the choices made in previous studies (Kratzert et al., 2019)."

It would be therefore arguable that the most effective hidden unit number would be always much higher than 64 since Lees et al. (2021) reported that this rule did not hold true in their work.

Please note that the preprint version of Lees et al. (2021) was published after Kratzert et al. (2019) and available at the time of preparation of our work. We based explicitly our choice of cell state size on this specific study, which was (at the time) the most recent study.

**2.3. Design of experiments**

RC: **It would be interesting (and useful) to know whether there is value in clustering catchments prior to training models, and if so whether we could find correlations between different hyperparameters (e.g., sequence length, cell state dimension) and hydrological regime (the former is a more interesting question than the latter, in my opinion). The way to test this is simple – you separately (and fully) hypertune each model. For example, if you want to test the clustering strategy described in lines 120-125, you would hypertune models separately for each catchment group (considering all of the important LSTM hyperparameters), and as a benchmark you would hypertune a model for all of the**

catchments combined. Then the results would be directly comparable. After that, you could look at whether there was any relationship between hydrological regime and the "optimal" (hypertuning is never actually optimal) sequence length for that cluster. If you really wanted to train single-basin models (which I suggest you should not do), then these need to be separately (and fully) hypertuned for each basin.

AR:  We appreciate the detailed description of the suggested design of experiments (DOE) and we acknowledge that it is thorough. We recognize that the reviewer identifies the following limitations for our DOE with respect to our research questions:

Limitation 1  In the first research question, the proposed hypothesis is that the effective size of lookback and regime of catchments are correlated. In order for the hypothesis to be valid, all lookbacks should have been tested for all regimes as well as the entire sample ($B_1$). This has been done only in local training and not for regional LSTMs.

Limitation 2  In the second research question, the hypothesis is that training less but hydrologically homogeneous catchments is more effective than training more catchments that are hydrologically heterogeneous. In our regional experiments, we have used lookbacks that were concluded at the local scale. Therefore, we might have not taken the most effective lookback for regional models when answering to the second research question.

Limitation 3  In the reviewer's opinion, the number of hidden units should have been varied along with the lookback size.

We plan to revise our DOE as follows:

Revision 1  For all regional — either per regime or per sample — models, we will test all the lookbacks tested for local models. This revision will deal with both Limitation 1 and Limitation 2.

In making this revision, we will consider 64 hidden units — this is necessary since in all local LSTMs 64 hidden units is used.

Revision 2  (Our answer to Limitation 3 has been provided above in Subsection 2.2. This revision is therefore provisional — it might be considered or not.) We will repeat Revision 1 using 256 hidden units. This size was concluded as most effective in Kratzert et al. (2019).

**2.4. Interest of local models**

RC:  I wonder why we are training local models. There is no situation where we would ever use a model trained on a single catchment for any real-world purpose. Additionally, the behavior of the LSTM is fundamentally and qualitatively different when trained on one catchment vs. many, which means that we cannot learn anything general or useful from locally trained models. If there was a specific hypothesis that we wanted to test that required training local models, then this might make sense, but I do not believe this is the case here – we could ask the question about appropriate sequence length on hydrologically grouped models, and asking the question this way would give us a more useful answer. Just a note: Kratzert et al. (in all papers after their 2018 paper) trained single-basin models only to make the point that this is not an appropriate thing to do.

AR:  We understand and share the reviewer's interest in using universal regional models. We also acknowledge the work of Kratzert et al. in developing regional models and their potential in transferring knowledge from

a number of known situations (gauged catchments) to unknown cases (ungauged catchments). There are however a number of reasons for which we disagree with the opinion that, in general, there is no interest in training local models, and, in particular, it has been done out of the purpose of the paper. These reasons from both general and particular perspectives are discussed in the following paragraphs.

General - 1    Compared to local models trained on individual catchments, regional models are expensive to develop and maintain — the slightest change in their setup necessitates re-training a huge model, even if we do not start from scratch. We agree that such computational expenses would not be a worry for Deep Learning researchers having access to the latest generation of GPUs. This would not be however the case for operational bodies from which real world applications arise. In particular, the interest of many of these operational entities (in France, but presumably elsewhere) is limited to a single (or very few) water course(s).

General - 2    The issue of computational cost of regional training is tied to another question: availability of national databases, which is often not the case — at least to public. In France, the SAFRAN data base used in this study has been made available only to very few research institutes. That is why we believe that our data set constitutes one of the novelty aspects of the paper noting that most of previous studies have been conducted using a single publicly available data set.

We therefore find it idealistic to think that none of these difficulties — i.e. availability of GPU resources and national data sets — will never emerge in practical applications.

General - 3    There are real world cases of catchment for which it would not be appropriate to use universal models. Dam influenced catchments are one of these cases. These catchments do not have natural responses and their non natural response is "unique" to them. We showed in our paper that in, even highly, influenced catchments, LSTM was able to learn the non natural response rule in these catchments. Such rules do not however constitute "transferable" knowledge since for each catchment is a function of the rule of dam, which itself depends on the dam purpose (flood control, irrigation, hydroelectric, ...), and the natural response rule of the catchment.

Paper - 1    In the first research question a general hypothesis is made — there is a link between the LSTM's lookback size and hydrological regime of catchments and it has not been specified that it would hold true solely for regional LSTMs. We believe that this hypothesis needed to be investigated at both scales in order to delimit its boundaries of validity.

Paper - 2    Local trainings in this paper are not carried out in the same way as any of the previous studies. Therefore, it would be arguable to immediately rule them out based on previous studies — having a basically different training approach — and without further investigation. Indeed, we used 10 years of data for validation, 10 other years for evaluation, and the size of train data in our sample varied between 10 and 40 years. While, for instance, in Kratzert et al. (2018) the local LSTM is trained using only 15 years of data and no data is used for validation. Or in Lees et al. (2021), although a validation period is taken into account, it is limited to 5 years and the length of train data does not go beyond 11 years while in our sample we have catchments with train data as long 40 years. (This point has been addressed in the paper, please see Section 5.2 of the preprint version). Also, train duration in our work was not imposed as a predefined fixed number of epochs. Furthermore, the GR4J model benefits from a particular feature — dealing with water gain/loss — making a clear distinction between the "baseline performances" in our work compared to previous studies. Taken all together, as we indicated in the conclusions of the paper (Line 507), we do not agree that regional LSTMs outperform, in any cases, local LSTMs

and all conceptual models. In particular, our results show that when train data are abundant, validation data are also present and sufficiently long, and train duration is not strictly imposed, local LSTMs could catch up with regional LSTM's level.

We find it thus arguable to state that regional LSTMs should be taken as the default choice in all cases of data and application and for all purposes.

**3. MINOR COMMENTS**

**3.1. S2 architecture**

RC: **The S2 architecture (stacked LSTMs) is interesting, but not related to either of the hypotheses of the study. What was the motivation for testing this and how does it relate to the questions that were motivated in the introduction? I'm not saying to remove it, just give us some reasoning or motivation. Also, when the "complexity" of this model is discussed, you might give us the number of free parameters so that we can get a sense of what the differences are.**

AR: Thank you for this interesting inquiry. The original intent behind studying the S2 architecture was to act on classical instructions that are given for training any Deep Learning models: prevent underfitting. We wanted to see if vertically stacking LSTMs could bring immediately a better performance thanks to a better hierarchical feature learning — in the same spirit of operating successive convolutions in a Convolutional Neural Network if we could "metaphorically" think that what LSTM looks for in time is comparable to what a CNN detects in space. But, we did not observe any instant improvement. There were thus two speculations: either, we still underfit a lot, or, the stacked setup would not basically help. Finding a definite answer to the question of "still underfitting" required an unmanageable amount of work — stacking more LSTMs and increasing hidden units until we overfit and repeating all local experiments at every step — and after all such effort, there was still the risk that "the stacked setup would not basically help". We therefore chose to assume that it was the second hypothesis that held true — stacking LSTMs vertically would not bring significant performance improvement. Thus, we ruled out the S2 structure.

Regarding the term "complexity" in the context of Neural Networks (NNs), the number of trainable parameters might not be a fully representative measure of model complexity in NNs since they are usually over parametrized. Indeed, one could find two different definitions of model complexity in the literature: 1) model expressive capacity (Bengio and Delalleau, 2011; Poggio et al., 2017) or 2) model usable capacity (Novak et al., 2018; Hanin and Rolnick, 2019). According to these definitions, different contributing factors are identified. For instance, choice of activation function, optimization algorithm as well as model size, which could be measured by the number of learnable parameters, the width, or the depth of the network. What we meant by the term "complex" in the paper was in terms of the depth of the network (its hierarchical representation).

We therefore completely agree with the reviewer that it should be specified what measure we are referring to when using the term "more complex" — in particular, when the S2 architecture involves less learnable parameters (13473) than the S1 (18497). We will update this sentence in the new version.

**3.2. Purpose of validation set (Line 192)**

RC: 💬 *The validation set is intended to be used for finding the best weights and biases during training and control overfitting.* **I think this is just a typo.**

AR: From this comment, we would infer that our hyperparameter tuning methodology is misunderstood — there exists no typo in the marked sentence. As we explained above, in our response to one of the earlier comments of the reviewer, we based our choices of batch size and model architecture on the results obtained using evaluation data and not validation data. Please refer to Subsection 2.1 of the present document.

RC: **Validation data is used to help find the best 💬hyperparameters and control overfitting (it is explicitly \*not\* used to help tune weights and biases, except through early stopping).**

AR: There is no typo/error in this phrase, neither. There is a missing explanation about how we chose model's best parameters. We did this in the same way as the `restore_best_weights` works in Keras. During training, for each epoch, we stored model parameters at the end of the epoch along with its validation loss. In the end of training, we identified the epoch corresponding to the best validation loss (smallest MSE). We then loaded the model with the parameters corresponding to this epoch and used this specific update of the model to make predictions on evaluation period (box plots with solid edges and darker colors in the paper's figures), as well as train + validation period (box plots with dashed edges and brighter colors in the paper's figures). Validation data are thus explicitly (or implicitly, depending on your definition in the current context) used to conclude on the best model parameters.

We will include this missing explanation in the revised version and we thank you for noticing this.

**3.3. Catchments with very short train data (Line 201)**

RC: **💬 What remains constitutes the train period (P1) the length of which varies between 1 year to 40 years in the FR sample.** **It is a little concerning to have different sized training data records per catchment, especially if some catchments only have 1 year of training data. This is \*especially\* problematic if we are looking at differences between what data is required to train in different types of catchments.**

AR: Please note that "all" catchments with less than 10 years of train data were excluded from our analyses at a very early stage. Please refer to Section 4.1 (Line 337) of the paper.

Please also note that none of the conclusions reported in our paper is based on the results obtained from such instances.

**3.4. Training duration (Line 180)**

RC: **In line 180 is reads like you are doing sequence-to-one prediction, however in line 259 you say that you are using a patience of 50 epochs with a maximum of 500 epochs. Typically you only need this many epochs if you are doing sequence-to-sequence training. Regardless, the number of epochs used by previous studies was in the range of 20-50. Have you found that more epochs help (we looked at this carefully in previous studies), or is there something else about your model that is different from previously published work?**

AR: Thank you for this interesting inquiry. One reason that we opted for the early stopping algorithm was that it would not impose to all catchments/simulations a wall condition — the same predefined non traversable training duration. Instead, it allows the model to continue to learn as long as its performance (on the validation set) is improving.

The so large numbers that we considered for these two parameters were meant to provide a free boundary condition. This would give the model the freedom to learn as long as it needed (unknown to us) without being stopped too early — due to either a too little patience or a too small maximum number of epochs.

This feature was advantageous to us since our data set was novel and had not been used in any of the previous studies notably that it contained catchments with very long sequences (up to 40 years).

The training duration — the epoch at which training has stopped — is available for all of our simulations and will be provided upon request.

**3.5. Static attributes (Line 291)**

RC: **This is a pretty small list of catchment attributes. Given that catchment attributes are available globally (e.g., HydroAtlas), and this will directly influence the generalizability of a model, why did we use such a limited set of attributes here?**

AR: Thank you for this relevant question. We agree with the reviewer that augmenting catchment specific data helps make local to regional mapping more effective, which in return reduces the generalization error. We believe however that not considering plenty of static attributes would be concerning if the model is being applied to ungauged catchments, which is not the case in our paper. We therefore took the classical static descriptors often used in previous regionalization studies in the French context (Oudin et al., 2008).

**3.6. Naming strategy**

RC: **In general, naming experiments with non-descriptive names like R1, R2. P1, etc. makes the paper more difficult to read than is necessary. This means that the reader must always refer back to the text in order to understand each figure. This can be solved simply by naming each of the models/experiments/datasets with descriptive names.**

AR: Thank you for this suggestion. Name revision will be considered to address this issue in all such instances.

**REFERENCES**

Y. Bengio and O. Delalleau. On the expressive power of deep architectures. In *International conference on algorithmic learning theory*, pages 18–36. Springer, 2011.

B. Hanin and D. Rolnick. Complexity of linear regions in deep networks. In *International Conference on Machine Learning*, pages 2596–2604. PMLR, 2019.

F. Kratzert, D. Klotz, C. Brenner, K. Schulz, and M. Herrnegger. Rainfall–runoff modelling using long short-term memory (lstm) networks. *Hydrology and Earth System Sciences*, 22(11):6005–6022, 2018. 10.5194/hess-2021-51110.5194/hess-22-6005-2018. URL https://hess.copernicus.org/articles/22/6005/2018/.

F. Kratzert, D. Klotz, G. Shalev, G. Klambauer, S. Hochreiter, and G. Nearing. Towards learning universal, regional, and local hydrological behaviors via machine learning applied to large-sample datasets. *Hydrology and Earth System Sciences*, 23(12):5089–5110, 2019. 10.5194/hess-2021-51110.5194/hess-23-5089-2019. URL https://hess.copernicus.org/articles/23/5089/2019/.

T. Lees, M. Buechel, B. Anderson, L. Slater, S. Reece, G. Coxon, and S. J. Dadson. Benchmarking data-driven rainfall–runoff models in great britain: a comparison of long short-term memory (lstm)-based models with four lumped conceptual models. *Hydrology and Earth System Sciences*, 25(10):5517–5534, 2021. 10.5194/hess-2021-51110.5194/hess-25-5517-2021. URL https://hess.copernicus.org/articles/25/5517/2021/.

R. Novak, Y. Bahri, D. A Abolafia, J. Pennington, and J. Sohl-Dickstein. Sensitivity and generalization in neural networks: an empirical study. *arXiv preprint arXiv:1802.08760*, 2018.

L. Oudin, V. Andréassian, C. Perrin, C. Michel, and N. Le Moine. Spatial proximity, physical similarity, regression and ungaged catchments: A comparison of regionalization approaches based on 913 french catchments. *Water Resources Research*, 44(3), 2008.

T. Poggio, K. Kawaguchi, Q.i Liao, B. Miranda, L. Rosasco, X. Boix, J. Hidary, and H. Mhaskar. Theory of deep learning iii: explaining the non-overfitting puzzle. *arXiv preprint arXiv:1801.00173*, 2017.

---

## Author Response (AR1)

**Author Response to Reviews of**

**How can regime characteristics of catchments help in training of local and regional LSTM based runoff models?**

Reyhaneh Hashemi et al.

*HESS,* `doi:10.5194/hess-2021-511`

RC: Reviewer Comment   AR: Author Response   RV: Revision

Dear Editor, Dear Referees,

We would like to invite you to find in the present document a summary of the major changes we have made to the paper (§1), our response to the report provided by Editor Efrat Morin (§2), our point by point response to the review made by Referee John Quilty (§3), and our point by point response to Anonymous Referee #2's review (§4).

Kind regards,
Authors

**1.   SUMMARY OF THE MAJOR CHANGES**

In the revised version, we have made the following major changes:

1. Sample — we have refined the initial sample by excluding:

   (a) the catchments with less than 30 years of full discharge record,

   (b) the influenced catchments with a degree of influence[1] greater than or equal to 0.1 ($d_i \geq 0.1$).

   The reduced sample that we have used for the revised manuscript has 361 catchments. We would also like to note that we do not have any more such an HP sample since the hyperparameter tuning approach has been fundamentally revised — according to Anonymous Referee #2's review.

2. Neural network architecture — we have dropped the S2 architecture of the old manuscript, which had 2 LSTM layers. In the revised manuscript, we have used the S1 architecture (from the old manuscript) for all LSTM experiments.

3. Learning rate — we have changed our learning rate to 0.0001. This is the learning rate that previous studies have reported (Kratzert et al., 2018; Lees et al., 2021).

4. Tuning hyperparameters — we have made the following changes:

   (a) in the revised manuscript, we have not tuned the "number of LSTM layers" and "batch size",
* * *
[1]The $d_i$ variable is defined in lines 95-99 of the old manuscript.

(b) as suggested by Anonymous Referee #2, we have included the "number of hidden units" in the tuning hyperparameters along with lookback. Following the previous studies, we have also added "dropout rate" to the hyperparameters to be tuned.

In the revised manuscript, we have thus considered six variations of lookback — $30, 60, 90, 180, 365, 730$ [days] — and 3 variations for hidden unit size — $64, 128, 256$ — and 3 dropout rates — $0.0, 0.2, 0.4$.

5. Hyperparameter tuning approach — we have performed a full "catchment wise" hyperparameter tuning for LSTMs trained on individual catchments and a full "model wise" hyperparameter tuning for LSTMs trained on a group of catchments, as proposed by Anonymous Referee #2. We therefore performed 54 hyperparameter tunings for each of the LSTMs found in the revised manuscript.

6. Approach to selection of the best hyperparameter set — we have investigated a new approach for group trained LSTMs. (Please see §3.5 of the revised manuscript.)

7. Static attributes — following the recommendation from Anonymous Referee #2, we have included four new attributes — mean daily solid precipitation, mean daily solid precipitation, mean daily potential evapotranspiration, and median altitude — in the static inputs of group trained LSTMs.

8. Research questions of the paper are revisited. Please see Introduction of the revised manuscript.

9. The title of the paper is changed to "How can we benefit from regime information to make use of LSTM runoff models more effectively?"

**2. EDITOR Efrat Morin**

Dear authors
We had review reports from two reviewers. There are a few major concerns that were raised mainly by reviewer #2, mostly focused on hypertuning. I would like to emphasize two points:
- The hypertuning should include more important LSTM parameters, including the cell state dimension, the sequence length (lookback), and others.
- For a fair comparison, hypertuning has to be done separately for each catchment group (including the group of all catchments). The hyperparameters found for the single catchments do not necessarily work for the catchment groups. If you decide to submit a revised paper, please address the above and other comments of the reviewers.

AR: We would like to thank you for the provided report and your conclusion, with which we fully agree. In the revised manuscript, we have fully followed the approach suggested by Anonymous Referee #2 and have totally revised our hyperparameter tuning approach. We have included "hidden unit size" in our tuning hyperparameters. Please see §4.2.1 and §4.2.3 for further details.

**3. REFEREE John Quilty**

**3.1. General comments**

RC: This paper carefully studies long short-term memory networks (LSTM) for rainfall-runoff prediction, using a large-sample of catchments in France. The key focus is on exploring local and regional models as well as the impact of the 'lookback' period, an important hyper-parameter of LSTM, with respect to predictive performance and physical understanding of the model results. The authors include well-thought out experiments to identify the impact of the lookback period and cases where local and regional LSTM models are best suited. The authors also benchmark LSTM with GR4J, due to its useful ability to capture ground water exchanges with aquifers and/or between catchments.

The authors spend a considerable amount of effort on tying the performance of LSTM, locally and regionally, to a physical understanding of the results. Some examples include the comparison between local and regional LSTM models with GR4J in terms of a water balance exercise in §5.3 as well as the ability of LSTM to predict runoff in controlled catchments at a higher degree of accuracy than GR4J (in §5.4). This paper also presents findings (e.g., LSTM does not necessarily outperform simple conceptual rainfall-runoff models) that are counter to other recent studies on LSTM (Gauch et al., 2021; Kratzert et al., 2019; Lees et al., 2021); in all such cases, the authors take the time to carefully describe potential reasons for these differences. Overall, this paper is very strong and I could not find much to criticize. The methodology seems correct. The figures are very nice and easy to interpret and I did not find any of the content, tables, or figures to be superfluous.

I suspect this paper will be very useful to other researchers interested in exploiting the generality of machine learning for hydrological modelling and rainfall-runoff prediction, in particular. I think the paper only needs some very minor corrections and some additional brief explanations (as noted below). Afterwards, the paper could be published.

AR: We would like to thank you for reading our manuscript carefully and with interest and for your encouraging comments. We are very pleased to read that you find our paper useful. We would like to invite you to find our

point by point response[1] to your comments in the following subsections.

**3.2. Specific comments**

**3.2.1 Line 32**

RC: **Does LSTM also help mitigate against exploding gradients? If so, this would be good to mention as well.**

AR: Thank you for this relevant question. The answer is, yes. This is because vanishing and exploding gradients both result from the same mathematical challenge when optimizing neural networks (NNs) with very high non linearities, although the latter case (exploding gradients) is less frequent (Goodfellow et al., 2016). A full description of how LSTM overcomes both vanishing and exploding gradients is given in Hochreiter and Schmidhuber (1997). To put it briefly and in very simple words, in deeply nested NNs, such as (vanilla) RNNs when the length of processing sequence ($T$) becomes large, it happens that a factor — which in the problematic case is not close to an absolute value of 1 — gets multiplied by itself over and over — $T$ times — due to the chain rule of the calculus. Therefore, the result will either exponentially shrink — if the factor is initially < 1 — or exponentially grow — if the factor is initially > 1 — and this is where the vanishing or exploding gradient issue arises. LSTM is designed to establish derivatives that neither vanish nor explode.

RV: Following the reviewer's suggestion, we have included this explanation in the revised manuscript (§3.1). We have also rewritten the whole section on LSTM's principles to make it more clear and tractable. We have removed unclear and confusing explanations and have given practical information on computation of different variables of the forward pass in an LSTM cell.

**3.2.2 Figure 1**

RC: **It would be good to include a description of the acronyms HP and FR in the figure caption (since it is unclear what these acronyms represent).**

AR: Thank you for this comment and we agree with you.

RV: Following your suggestion and the point made by the Anonymous Referee #2 on our naming strategy, we have revised all instances of acronym/letter based names — all such instances are replaced by descriptive names.

**3.2.3 Grammatical corrections**

RC: **For the most part, the paper is well-written but there are a number of grammatical errors. I stopped correcting such errors around line 154. I recommend that a very carefully read through the paper be completed before re-submission.**

AR: Thank you for spotting these grammatical errors. We agree with you about the section containing Line 154 and its surrounding. We found it also a bit stiff and wordy.

RV: We have thus totally rewritten this section to make it more concise, clear, and useful. We hope it reads well now in the revised manuscript (§3.1). We have also proofread the entire paper to check for any further errors.
* * *
[1] All line and figure mentions found in the title of the subsections of this section regard the old manuscript.

**3.2.4  Line 165**

**RC:**  **The sentence on line 165 can be moved to the last sentence of the same sub-section. A short sentence, 'The main equations used in LSTM are as follows (Ref, XXX):' can be used in it's place.**

AR:  Thank you for this suggestion.

RV:  We have rewritten this subsection, as mentioned in the above RV.

**3.2.5  Lines 205-206**

**RC:**  **Is this sort of standardization the most appropriate for LSTM? Since sigmoid and tanh activation functions are used, should not the data be scaled to [0,1] or [-1,1] as these ranges match the output ranges of the (previously mentioned) activation functions? Perhaps others have adopted the form of standardization adopted here, if so, can the authors indicate this?**

AR:  Thank you for this interesting question. LeCun et al. (2012) explain that why centering the input data around 0 and scaling them by the standard deviation is typically a good idea and usually makes gradient descent converge faster. Besides, we could not in principle benefit from a [0, 1] scaling since the temperature feature might include negative values.

The interesting point of standardization comes when we investigate how the derivative of different activation functions changes with respect to the range of input data. Please note that here we are talking about the activation function for the hidden layers and not the last layer, which is given by the type of the problem — for instance, Softmax for multi class classification, Sigmoid for binary classification, and Identity for regression. Looking at Fig. 1 (of the present document), it turns out that the Sigmoid ($\sigma(x)$) and tanh functions suffer from a problem — their derivative gets saturated very quickly. By the term "saturation", we mean that their derivative approaches very quickly to zero indicating that weights can not get updated effectively at these points thus the NN can not learn effectively. We observe this problem almost everywhere except in the small region in the middle centered around 0 where the derivative is the most dynamic. Therefore, having the inputs centered around 0 with a variance of 1 would also help fall in the useful area of these functions. Please note that even in their dynamic region the derivatives are small and could bring about the vanishing gradient problem in NNs with high non linearities.

Now, you might ask why not simply using an activation function that does not have vanishing gradients, for instance ReLu? The answer is that the ReLu activation function proved to be typically a more appropriate default choice — if we are allowed to use it. For LSTM, there is a specific reason for which we need to stick with the Sigmoid function in gates, despite the mentioned disadvantages. Indeed, the it plays a "gate" role — a function granting us a value between 0 and 1 — and it is not possible to replace it by ReLu or any other activation functions not having this output range.

RV:  We have included a summary of this answer in §3.2 of the revised manuscript, where we have also referred to Kratzert et al. (2018), who standardized their data using the mean and standard deviation of the training data.

**3.2.6  Adam algorithm**

**RC:**  **What were the hyper-parameters (alpha, beta_1, beta_2) set to in the Adam algorithm?**

AR:  Except for learning rate ($\alpha$) that we have set to $0.0001$ in the revised manuscript, we have kept all other arguments, including $\beta_1$ and $\beta_2$ ($L^1$ and $L^2$ norms), at their default values in Keras (Chollet et al., 2015):

[Figure]

(a) $\sigma(x)$ vs. $\dfrac{d[\sigma(x)]}{d(x)}$     (b) $\tanh(x)$ vs. $\dfrac{d[\tanh(x)]}{d(x)}$

Figure 1: Sensitivity of the derivative of the Sigmoid and $\tanh$ functions to the range of input data.

```
tf.keras.optimizers.Adam(
    learning_rate =0.0001,
    beta_1=0.9,
    beta_2=0.999,
    epsilon=1e-07,
    amsgrad=False,
    name="Adam",
    )
```

RV:   In the revised manuscript, we have included a phrase indicating this setting (§3.5).

**3.2.7   Equation 17**

RC:   **what does epsilon represent?**

AR:   Thank you for noticing this — we had forgotten to indicate that Kratzert et al. (2019) added this term ($\epsilon$) to the denominator in the equation of $\text{NSE}^*$ so that the loss function would not explode when $s$ was very close to 0 (catchments with very small discharge variance).

RV:   We have updated the text to include this explanation (§3.5).

**3.3.   Technical corrections**

RC:   **Lines 16, 35, 57, 61, 73, 135, 148-150, 154, Figure 6**

AR:   Thank you for spotting these errors.

RV:   The manuscript is (almost) rewritten. We tried to avoid such errors in the new manuscript and hope that such errors have not accrued in the new version.

**4. ANONYMOUS REFEREE #2**

AR:   We would like to thank you, Anonymous Referee #2, for your review and constructive thoughts, which have greatly improved our paper. You were critical of the design of our experiments and hyperparameter tuning approach and had identified technical issues warranting a major revision of these two aspects. We agree with you. We have made all the revisions you had suggested. We have updated our point by point response to your individual comments according to these revisions and would like to invite you to find it below.

**4.1. Summary of review**

RC:   **This paper addresses two research questions related to the use of LSTMs for rainfall-runoff modeling: (1) Does appropriate sequence length depend on hydrological regime, and (2) should LSTM training be done on hydrologically similar basins?**
**To state my opinion up front, I have run similar experiments (unpublished) and found results that are qualitatively different than what are reported here. There are several technical issues in this paper (overall, the methodology is not appropriate for testing the stated hypotheses), and it might be worth addressing those before we look carefully at the results.**

AR:   We thank the reviewer for their interest in performing similar experiments. We would be happy to engage in an ongoing dialogue with the reviewer about the details of their experiments since without further information, in particular, on the hydro-geo-climatic context of their data, it would be hard to provide a definite explanation. Indeed, such discrepancies could be investigated at different levels. At the highest level, we would conjecture that the reason lies in definition of the homogeneity component. We believe that the hydrological similarity rule — i.e., the regime classification — is a crucial question. Given the term "similar experiments", we could think of the following two cases.

Case 1   The exact same classification is applied to a sample in a non French context. In this case, we would be afraid that the exact same rule would not be immediately applicable to other climatic contexts. The following elaboration on our classification approach aims to underline how it is intensely context dependent — in terms of number of classes, criteria, and thresholds.

As a property of the French context, we knew in advance that there existed five main regime patterns, which we named Uniform, Mediterranean, Oceanic, Nival Pluvial, and Nival. Therefore, any catchment in our sample could be classified in one of the five categories. Using the fact sheets available at https://webgr.inrae.fr/activites/base-de-donnees/, we tried to identify hydro-geo-climatic signals that could reflect different features of all five patterns. The decision feature(s) — based on which we could distinguish one pattern from the others — was (were) not the same for all regimes. For instance, we were observing that the minimum temperature attribute alone was able to detect the Nival pattern. While, in catchments with known water ground effects, it was certainly a criterion on discharge that was doing this. In the same spirit of a decision tree algorithm, but at a human level, we concluded that the mean annual discharge, total precipitation, and temperature signals would be the most useful signals to exploit to identify the distinctive attribute(s) in each class. After a number of trial and errors on different properties of these signals, such as the number, magnitude, and time of occurrence of their global/local peaks, we identified our classification attributes — $IQ$, $IP$, and $T_{\min}$ — and their thresholds. Therefore, inferring the similarity rule for any other climatic context warrants a redefinition of different elements in this analysis.

Case 2   Another classification (concluded at an AI or a human level) is applied to data belonging to a non French context. In this case, getting different results would not be surprising — since a different rule

involves different decision attributes and criteria. The question in this case would be rather the extent to which we could compare the results obtained from two different classifications.

To conclude on this point, we believe that such cross study comparisons warrant caution, since the imposed similarity rule will not be identical between the studies. That is why we would like to emphasis and acknowledge that research questions established on a subjective component, such as regime classification in our paper, will always make the corresponding conclusions subject to that component.

RC:   **My overall recommendation is to revise the experiment as suggested in one of the comments below. The experimental design that is appropriate to test the (two) hypotheses outlined here is very simple (but somewhat computationally expensive). If the authors were to find similar results using a more appropriate experiment, this would be an interesting study.**

RV:   We provide the details of our revision later in §4.2.3 where the reviewer details their suggested design of experiments.

**4.2. Comments**

**4.2.1   Hyperparameter tuning**

RC:   **Hyperparameter tuning was done on LSTMs trained on individual basins. LSTMs trained on individual basins behave fundamentally differently than LSTMs trained on multiple basins, which means that lessons learned from hypertuning on individual basins do not translate to multiple-basin models.**

RV:   We have totally revised our hyperparameter tuning approach following the methodology suggested by the reviewer. We have defined 54 hyperparameter tunings, where 54 reflects the number of all possible combinations of the three considered hyperparameters — LSTM sequence length with 6 variations, hidden unit number with 3 variations, and dropout rate with 3 variations; that is $6 \times 3 \times 3$. For "each and every one" of the paper's LSTMs — either trained on individual or a group of catchments — we have performed these 54 tunings. Please see §3.2 and §3.4 of the revised manuscript.

RC:   **Additionally, 15 catchments is not enough for robust hypertuning – we would need to perform hyperparameter tuning on the full (evaluation) dataset (although see a later comment – the experimental design needs to be changed fundamentally). Also, notice that the only portions of the "hypertuning" that were actually used for the other experiments in this paper were (1) discarding the S2 model architecture, and (2) batch size.**

RV:   Please see the above RV on our new hyperparameter tuning approach.

**4.2.2   Number of hidden units in the LSTM layer**

RC:   **There is strong relationship between the dimension of the cell state and the sequence length, and also between the cell state dimension and the ability of the model to generalize (Kratzert et al. (2019) shows how the model uses the cell state to map catchment similarity). This parameter was not included in the hyperparameter tuning, and it was also not considered in the experimental design. 64 cell states is smaller than used by most of the previously published work. The hypotheses that are tested here are about the ability of the model to generalize and about memory timescales, both of which are directly controlled by the cell state (more cell states means more ability to have different memory timescales for different hydrological regimes).**

AR: We thank the reviewer for this relevant and constructive comment regarding the number of hidden units in the LSTM layer.

RV: In addition to 64, we have included two larger hidden unit sizes — 128 and 256 — in our hyperparameter tuning.

**4.2.3 Design of experiments**

RC: **It would be interesting (and useful) to know whether there is value in clustering catchments prior to training models, and if so whether we could find correlations between different hyperparameters (e.g., sequence length, cell state dimension) and hydrological regime (the former is a more interesting question than the latter, in my opinion). The way to test this is simple – you separately (and fully) hypertune each model. For example, if you want to test the clustering strategy described in lines 120-125, you would hypertune models separately for each catchment group (considering all of the important LSTM hyperparameters), and as a benchmark you would hypertune a model for all of the catchments combined. Then the results would be directly comparable. After that, you could look at whether there was any relationship between hydrological regime and the "optimal" (hypertuning is never actually optimal) sequence length for that cluster. If you really wanted to train single-basin models (which I suggest you should not do), then these need to be separately (and fully) hypertuned for each basin.**

AR: We appreciate the detailed description of the suggested design of experiments (DOE) and we acknowledge that it is thorough. We recognize that the reviewer identifies the following limitations for our old DOE with respect to our research questions:

Limitation 1   We had proposed a hypothesis: the effective size of lookback and regime of catchments are correlated. In order for the hypothesis to be valid, all lookbacks should have been tested for all regimes as well as the entire sample. In the old manuscript, this had been done only in local training and not for regional LSTMs.

Limitation 2   The second hypothesis was that training less but hydrologically homogeneous catchments would be more effective than training more but hydrologically heterogeneous catchments. In our old regional experiments, we had used the lookbacks that were concluded at the local scale. Therefore, we might have not taken the most effective lookback for regional models when answering to the second research question.

Limitation 3   In the reviewer's opinion, the number of hidden units should have been varied along with the lookback size.

RV: We have revised our DOE as follows taking into account the reviewer's suggestions and the three identified limitations:

Revision 1   For all group trained LSTMs, we have tested all and the exact same lookbacks that have been tested for LSTMs trained on individual catchments. This revision has addressed Limitation 1 and 2.

Revision 2   In doing Revision 1, we have tested 3 different hidden units $\geq 64$ for all local and regional LSTMs to address Limitation 3.

**4.2.4 Interest of local models**

**RC:** **I wonder why we are training local models. There is no situation where we would ever use a model trained on a single catchment for any real-world purpose. Additionally, the behavior of the LSTM is fundamentally and qualitatively different when trained on one catchment vs. many, which means that we cannot learn anything general or useful from locally trained models. If there was a specific hypothesis that we wanted to test that required training local models, then this might make sense, but I do not believe this is the case here – we could ask the question about appropriate sequence length on hydrologically grouped models, and asking the question this way would give us a more useful answer. Just a note: Kratzert et al. (in all papers after their 2018 paper) trained single-basin models only to make the point that this is not an appropriate thing to do.**

**AR:** We agree with the reviewer and share their interest in using universal regional models. We also agree with the reviewer on the point that hyperparameters of local models are not optimal for regional LSTMs. That is why in the revised manuscript we have performed a full hyperparameter tuning for all models, as suggested by the reviewer. In the revised manuscript, we have kept the local LSTM to be able to study our first research question at the local scale as well. We have been also interested in benchmarking the LSTM against the GR4J model. We believe that having a local baseline would also help the reader to better interpret and understand different aspects of the regional results that we present, such as the performance gain in the passage from local to regional trainings.

**4.3. Minor comments**

**4.3.1 S2 Architecture**

**RC:** **The S2 architecture (stacked LSTMs) is interesting, but not related to either of the hypotheses of the study. What was the motivation for testing this and how does it relate to the questions that were motivated in the introduction? I'm not saying to remove it, just give us some reasoning or motivation. Also, when the "complexity" of this model is discussed, you might give us the number of free parameters so that we can get a sense of what the differences are.**

**AR:** Thank you for this interesting inquiry. The original intent behind studying the S2 architecture in the old manuscript was to act on classical instructions that are given for training any Deep Learning models: prevent underfitting. We wanted to see if vertically stacking LSTMs could immediately bring a better performance thanks to a better hierarchical learning — in the same spirit of operating successive convolutions in a Convolutional Neural Network, if we could "metaphorically" think that what LSTM looks for in time is comparable to what a CNN detects in space. But, we did not observe any instant improvement. There were thus two speculations: either, we still underfit a lot, or, the stacked setup would not basically help. Finding a definite answer to the question of "still underfitting" required an unmanageable amount of work — stacking more LSTMs and increasing hidden units until we overfit and repeating all local and regional experiments at every step — and after all such effort, there was still the risk that "the stacked setup would not basically help". We therefore chose to assume that it was the second hypothesis that held true — stacking LSTMs vertically would not bring significant performance improvement. Thus, we ruled out the S2 structure.

**RV:** We have excluded this architecture from the revised manuscript.

**4.3.2 Purpose of validation set (Line 192)**

**RC:** *The validation set is intended to be used for finding the best weights and biases during training and control overfitting.* **I think this is just a typo. Validation data is used to help find the best *hyperparameters* and control overfitting (it is explicitly \*not\* used to help tune weights and biases, except through early stopping).**

**RV:** In the revised manuscript, for all local and regional trainings, we have used the validation data to select the best hyperparameter set as well as in the Early Stopping algorithm.

**4.3.3 Catchments with very short train data (Line 201)**

**RC:** *What remains constitutes the train period (P1) the length of which varies between 1 year to 40 years in the FR sample.* **It is a little concerning to have different sized training data records per catchment, especially if some catchments only have 1 year of training data. This is \*especially\* problematic if we are looking at differences between what data is required to train in different types of catchments.**

**AR:** In the old manuscript all catchments with less than 10 years of training data had been excluded from the analyses at a very early stage.

**RV:** In the revised manuscript, we have excluded all catchments with less than 10 years of training data from the study.

**4.3.4 Training duration (Line 180)**

**RC:** **In line 180 is reads like you are doing sequence-to-one prediction, however in line 259 you say that you are using a patience of 50 epochs with a maximum of 500 epochs. Typically you only need this many epochs if you are doing sequence-to-sequence training. Regardless, the number of epochs used by previous studies was in the range of 20-50. Have you found that more epochs help (we looked at this carefully in previous studies), or is there something else about your model that is different from previously published work?**

**AR:** Thank you for this interesting inquiry. One reason that we opted for the early stopping algorithm was that it would not impose to all catchments/simulations the same predefined non traversable training duration. Instead, it allows the model to continue to learn as long as its performance (on the validation set) is improving.

The so large numbers that we considered for these two parameters were meant to provide a free boundary for training duration. This would give the model the freedom to learn as long as it needed (unknown to us) without being stopped too early — due to either a too little patience or a too small maximum number of epochs.

This feature was advantageous to us since our data set was new and had not been used in any of the previous studies, notably that it included catchments with very long time series — sometimes up to 40 years for the training data.

**4.3.5   Static attributes (Line 291)**

RC:   **This is a pretty small list of catchment attributes. Given that catchment attributes are available globally (e.g., HydroAtlas), and this will directly influence the generalizability of a model, why did we use such a limited set of attributes here?**

AR:   Thank you for this relevant question. We agree with the reviewer. In the old manuscript, we thought that not considering plenty of static attributes would be concerning if the model was to apply to ungauged catchments, which was not the case in our paper. We therefore took the classical static descriptors often used in previous regionalization studies in the French context (Oudin et al., 2008).

RV:   In the revised manuscript, we have included four more attributes — mean daily solid precipitation, mean daily solid precipitation, mean daily potential evapotranspiration, and median altitude — in the static inputs of group trained LSTMs, giving in total 10 static features.

**4.3.6   Naming strategy**

RC:   **In general, naming experiments with non-descriptive names like R1, R2. P1, etc. makes the paper more difficult to read than is necessary. This means that the reader must always refer back to the text in order to understand each figure. This can be solved simply by naming each of the models/experiments/datasets with descriptive names.**

AR:   We fully agree with the reviewer and thank them for this constructive suggestion.

RV:   We have replaced all such instances by descriptive names as suggested by the reviewer. We have, for instance, replaced P1, P2, and P3 by training, validation, and test. We have chose SINGLE, REGIONAL, and HYBRID names for our LSTMs.

**REFERENCES**

F. Chollet et al. Keras, 2015. URL https://github.com/fchollet/keras.

M. Gauch, J. Mai, and J. Lin. The proper care and feeding of camels: How limited training data affects streamflow prediction. *Environmental Modelling Software*, 135:104926, 2021. ISSN 1364-8152. 10.5194/hess-2021-511https://doi.org/10.1016/j.envsoft.2020.104926. URL https://www.sciencedirect.com/science/article/pii/S136481522030983X.

I. Goodfellow, Y. Bengio, and A. Courville. *Deep Learning*. MIT Press, 2016.

S. Hochreiter and J. Schmidhuber. Long short-term memory. *Neural computation*, 9(8):1735–1780, 1997.

F. Kratzert, D. Klotz, C. Brenner, K. Schulz, and M. Herrnegger. Rainfall–runoff modelling using long short-term memory (lstm) networks. *Hydrology and Earth System Sciences*, 22(11):6005–6022, 2018. 10.5194/hess-2021-51110.5194/hess-22-6005-2018. URL https://hess.copernicus.org/articles/22/6005/2018/.

F. Kratzert, D. Klotz, G. Shalev, G. Klambauer, S. Hochreiter, and G. Nearing. Towards learning universal, regional, and local hydrological behaviors via machine learning applied to large-sample datasets. *Hydrology and Earth System Sciences*, 23(12):5089–5110, 2019. 10.5194/hess-2021-51110.5194/hess-23-5089-2019. URL https://hess.copernicus.org/articles/23/5089/2019/.

Y.-A. LeCun, L. Bottou, G.-B. Orr, and K.-R. Müller. Efficient backprop. In *Neural networks: Tricks of the trade*, pages 9–48. Springer, 2012.

T. Lees, M. Buechel, B. Anderson, L. Slater, S. Reece, G. Coxon, and S. J. Dadson. Benchmarking data-driven rainfall–runoff models in great britain: a comparison of long short-term memory (lstm)-based models with four lumped conceptual models. *Hydrology and Earth System Sciences*, 25(10):5517–5534, 2021. 10.5194/hess-2021-51110.5194/hess-25-5517-2021. URL https://hess.copernicus.org/articles/25/5517/2021/.

L. Oudin, V. Andréassian, C. Perrin, C. Michel, and N. Le Moine. Spatial proximity, physical similarity, regression and ungaged catchments: A comparison of regionalization approaches based on 913 french catchments. *Water Resources Research*, 44(3), 2008.

---

## Referee Report (RR1)

**Review hess-2021-511 R1**

**TITLE**

How can we benefit from regime information to make use of LSTM runoff models more effectively?

Formerly: How can regime characteristics of catchments help in training of local and regional LSTM-based runoff models?

**RECOMMENDATION**

Accept

**REVIEWER**

John Quilty

**GENERAL COMMENTS**

The authors have satisfactorily addressed my comments on their initial manuscript. I recommend the paper be published after correcting the minor issues noted below.

Thank you for the opportunity to review this interesting paper!

**MINOR COMMENTS**

All comments below refer to the track-changes version of the article.

1. A reference should be included in the sentence just before Equation 4. The source of all equations that were not developed by the authors in this paper should be clearly indicated.
2. Line (L) 250-251: "Also, since the temperature feature can take negative values, we can not in principle benefit from a [0, 1] scaling." What principle is being referred to? If a variable that takes negative values is scaled to [0, 1], it still (qualitatively) conveys the same information (as it did on the original scale), where the origin is shifted from 0 to some corresponding number in [0, 1].
3. Table 3: should the terms 'runoff index' and 'total precipitation index' in the first column be swapped with one another?
4. Equation 15: what was epsilon set to?
5. Grammatical issues: I noticed a few grammatical issues related to the authors' replies to my various comments from the initial review. I suspect there may be others. Again, I suggest the authors do another run through the manuscript to catch similar issues.

   a. L210: 'brief' instead of 'quick.
   b. L225: 'disregard' instead of 'throw away'.
   c. L233: remove 'of'.

---

## Author Response (AR2)

**Author Response to Reviews of**

**How can we benefit from regime information to make more effective use of LSTM runoff models?**

Reyhaneh Hashemi et al.
*HESS,* `doi:10.5194/hess-2021-511`

RC: Reviewer Comment  AR: Author Response   RV: Revision

Dear Editor, Dear Referees,

Thank you for the time you took to review our revised manuscript and for your reports.

We would like to invite you to find our point-by-point response to the reports provided by the referees John Quilty and Anonymous Referee #3 in sections 1 and 2 of the present document, respectively.

Kind regards,
Authors

**1. REFEREE: John Quilty**

**1.1. General comments**

RC: **The authors have satisfactorily addressed my comments on their initial manuscript. I recommend the paper be published after correcting the minor issues noted below.**

**Thank you for the opportunity to review this interesting paper!**

AR: We would like to thank you for reading our revised manuscript and your interest. Please find in the following paragraphs our point-by-point response to your comments or/and the modifications made in the second revision.

**1.2. Minor comments**

RC: **All comments below refer to the track-changes version of the article.**

**1.2.1 Reference for the equations of Section 3.1 — A primer in Long Short Term Memory (LSTM)**

RC: **A reference should be included in the sentence just before Equation 4. The source of all equations that were not developed by the authors in this paper should be clearly indicated.**

RV: Following the reviewer's comment, we have have included the following sentence in §3.1 of the revised manuscript: "Equations (4) to (9) given below are all from Goodfellow et al. (2016), with a slightly different notation.".

**1.2.2 Choice of standardization (Lines 250-251)**

RC: **"Also, since the temperature feature can take negative values, we can not in principle benefit from a [0, 1] scaling." What principle is being referred to? If a variable that takes negative values is scaled to [0, 1], it still (qualitatively) conveys the same information (as it did on the original scale), where the origin is shifted from 0 to some corresponding number in [0, 1].**

AR: Thank you for this relevant comment, you are right. The min–max normalization is still theoretically applicable, although, for the reasons mentioned in the first AR document, it is recommended to use standardization (the Z–score normalization).

RV: We have removed the referenced sentence from the manuscript.

**1.2.3 Table 3**

RC: **Should the terms 'runoff index' and 'total precipitation index' in the first column be swapped with one another?**

AR: Yes; thank you for spotting this mistake.

RV: We have done the swap in the second revision of the manuscript.

**1.2.4  epsilon in Equation 15**

RC:  **What was epsilon set to?**

AR:  Following Kratzert et al. (2019), we set epsilon ($\epsilon$) in Equation 15 to 0.1.

RV:  We have updated the paragraph just after Equation 15 as follows: "Following Kratzert et al. (2019), $\epsilon$ ($= 0.1$) is added to the denominator in equation NSE* [...]".

**1.2.5  Grammatical issues**

RC:  **I noticed a few grammatical issues related to the authors' replies to my various comments from the initial review. I suspect there may be others. Again, I suggest the authors do another run through the manuscript to catch similar issues.**

    a.  **L210: 'brief' instead of 'quick'.**

    b.  **L225: 'disregard' instead of 'throw away'.**

    c.  **L233: remove 'of'.**

AR:  Thank you for this suggestion. The new revised manuscript has been read and edited thoroughly and all issues related to style and register, or grammar, have been addressed.

**2. REFEREE: Anonymous Referee #3**

**2.1. Review summary**

RC: **In this paper entitled "How can we benefit from regime information to make use of LSTM runoff models more effectively?", Hashemi et al. developed long short term memory (LSTM) models to assess their capability for runoff modeling according to how long memory (lookback hyperparameter) depends on hydrological regimes (i.e. on the information existing up to annual time scale), how the models are trained (local, regional or "national"-scale training), and in the end, answer the question "what is the most effective way of using LSTM for making runoff predictions?" (quite a broad question).**
**This paper, which has undergone a number of modifications by the authors already, is overall very well written and organized, with clear objectives. This type of paper certainly deserves being brought to the hydrological community. I have a few concerns, though, that I think should be addressed before the paper be considered for final publication. They meet, to some extent, those already expressed previously by one reviewer. The authors will decide whether they can just use these comments below to modify the text or if additional trials are needed.**

AR: We would like to thank you for your review as well as your comments and suggestions. Please find below our point-by-point response to your individual comments.

**2.2. Main comment**

RC: **It would have been probably better to explore a little deeper the parameter space in my opinion (as emphasized by reviewer 2 previously). At least, should the paper be published, it is mandatory to explain why some important parameters were kept constant and what is the rationale behind this decision: otherwise, my feeling is it will not be of sufficient help to the readership and potential users to use this work as a support to develop their own models, for instance.**
**I am not saying the values of the parameters are not suited, but without any \*strong rationale\* (physical or anything else) supporting this choice, it is difficult, in the framework of ML/DL approaches, to justify the selection of just a few values of a limited number of hyperparameters.**

AR: Thank you for this relevant comment. In the following paragraphs, we explain for each hyperparameter individually why it has been tuned (or not) and why some certain exploring values (and not others) have been taken into account.

**2.2.1 Learning rate**

In the present paper, Adam is used as the optimization algorithm and its base learning rate is fixed to $10^{-4}$.

*Why did not we tune its (base) learning rate?* Adam (Kingma and Ba, 2014) is from the family of algorithms with *adaptive* learning rates and is considered to be a robust algorithm with respect to the choice of its hyperparameters, including its base learning rate (Goodfellow et al., 2016).

*Why did we fix it to $10^{-4}$?* A suitable learning rate value would give an asymptotic converging (optimization) learning curve and would not overshoot effective local minima (Bengio, 2012). Given these factors, we fixed Adam's basic learning rate to $10^{-4}$ and performed a post hoc examination of the (optimization) learning curves for the different models in the experiments we conducted that did not reveal any divergence of the training criteria due to a too high learning rate. The rate of $10^{-4}$, which is 10 times lower than Adam's default

[Figure]

Figure 1: [*Both figure and caption are from the textbook of Goodfellow et al. (2016).*] An illustration of the effect of early stopping. (Left)The solid contour lines indicate the contours of the negative log-likelihood. The dashed line indicates the trajectory taken by SGD beginning from the origin. Rather than stopping at the point $\omega^*$ that minimizes the cost, early stopping results in the trajectory stopping at an earlier point $\tilde{\omega}$. (Right)An illustration of the effect of $L^2$ regularization for comparison. The dashed circles indicate the contours of the $L^2$ penalty, which causes the minimum of the total cost to lie nearer the origin than the minimum of the unregularized cost.

base learning rate (in Keras), was selected to provide better steps with respect to local minima. Given this lower chosen learning rate, in order to ensure that full regime training had been provided and that the training criterion had sufficient time to decay, we did not impose a predetermined number of epochs, instead allowing the LSTM to continue to learn for as long as its performances improved in the validation data. Further, $10^{-4}$ is the chosen value in similar previous studies (Kratzert et al., 2018; Lees et al., 2021).

RV:   We have added a discussion including the elements provided in this answer to §3.4 of the revised manuscript.

**2.3. Dropout rate**

AR:   *Why did not we consider a larger variation for dropout?* The early stopping algorithm implemented in the paper already acts as a regularizer. Goodfellow et al. (2016) show how, in the case of a simple linear model with a quadratic error function and simple gradient descent early stopping is equivalent to $L^2$ regularization (see Figure 1 taken from Goodfellow et al. (2016)). Our results showed that the use of a second regularization strategy (dropout rates $> 0$) in conjunction with early stopping would not further improve performance (compared to the use of early stopping alone, i.e., dropout rate $= 0$). This is consistent with the results provided in previous studies by Kratzert et al. (2019) and Lees et al. (2021), where no other regularization (e.g., early stopping) is implemented and dropout rates $> 0$ give better results than dropout rate $= 0$. Given this, there was no point testing more dropout variations in our paper.

RV:   We have added a discussion including the elements provided in this answer to §3.4 of the revised manuscript.

**2.3.1 Batch size**

 **For instance, it is not clear why batch size was kept to 128.**

AR: This is a very interesting question; thank you.

*Why did not we tune batch size?* Bengio (2012) notes that the impact of the size of training batches ($B$) is mostly computational and that, theoretically, $B$ should mainly impact training times and convergence speeds, with no significant impact on test performance. That is, larger $B$s would speed up computation but need to encounter more samples in order to arrive at the same error since there are fewer updates per epoch, and vice versa for smaller $B$s.

*Why did we fix it to* $128$*?* Typical recommended batch sizes are powers of 2 (since they offer a better GPU runtime), ranging from 32 to 256 (Goodfellow et al., 2016). Very small batch sizes might require a lower learning rate to maintain stability due to the high variance in gradient estimates. Thus, the total runtime can increase significantly when more steps are required to 1) visit the entire sample, and 2) converge (because a lower learning rate is used). Our chosen learning rate—batch size ($10^{-4}$, 128) – gave a reasonable run time and adequate convergence and test performance.

RV: We have added a discussion including the elements provided in this answer to §3.4 of the revised manuscript.

**2.3.2 Hidden unit ($HU$) size**

RC: **Or why only 64, 128, 256 hidden units were eventually selected: not less, nothing in between? By the way, is there any specific reason for choosing log2 values? I don't think any numerical constraints would require this in the present context and gaps between successive values are large...**

AR: *Why* $\log 2$ *values? Why nothing in between?* Bengio (2012) offers an interesting discussion on the recommended exploration values for a hyperparameter the "Scale of values considered" paragraph of §3.3 of his paper. He explains that, instead of making a linear selection of intermediate value intervals (the values between the lower and upper bands, here 64 to 256), it is often much more useful to consider a linear or uniform sampling in the log domain — in the space of the logarithm of the hyperparameter. This is because the "ratio" between different values is often more important than their absolute difference when it comes to "the expected impact of the change". For this reason, Bengio (2012) states that exploring uniformly or regularly spaced values in the space of the logarithm of the numerical hyperparameter is typically to be preferred for positive valued numerical hyperparameters.

*Why not less?* Should the optimal $HU$ be lower than $64$, using a $HU$ of $64$ would not negatively impact generalization performance, it would simply require proportionally greater computation (Bengio, 2012).

RV: We have added a discussion including the elements provided in this answer to §3.4 of the revised manuscript.

**2.3.3 Sequence lengths greater than 2 years**

RC: **Also, I am wondering if it would have been interesting to use sequence lengths (lookback) up to, say, 4 years: I have not seen what the streamflow time series look like but for some of them with strong baseflow and high multi-annual variability (as visible in some regimes of fig.4), it might be possible that some useful information be still present further back in time (even more than 2 years), and that**

**the annual scale ("regime") does not necessarily contains all the useful information by itself (there have been quite an amount of works published in the past decade on the topic). Without that, it will be probably difficult to provide a meaningful answer to question Q4 about "[...] the most effective way of using LSTM for making runoff predictions", in my opinion...**

AR: Thank you for this relevant and interesting comment. We fully agree with you on this point and also believe that the sweet spot [1] for lookback could go beyond 2 [years], in particular, in Uniform catchments. Our paper's focus has just been to show that the (minimum) required lookback would vary depending on the catchment hydrologic characteristics. Question "Up to how many years?" is not addressed in the present paper, albeit very interesting.

Our goal to address Q4 has been to emphasis on the approach involved — group training + *local hyperparameter selection*. In that, we would like to note that local parameter search could be performed in a space of whatever size.

**2.4. Minor suggestions**

**2.4.1 Hysteresis versus memory length (Introduction Section — Line 52)**

RC: **I think that confusing the hysteretic behavior and the memory length of a catchment is not strictly speaking true: the first relates mainly to the lagged response to the input, the second to the time taken by the system to dissipate the information of the input.**

AR: Thank you for this comment.

RV: To prevent confusion, "hysteresis" and all of its derivatives have been removed throughout the manuscript. Instead, the "long term dynamics" or "temporal dynamics" terms have been used, depending on the context.

**2.4.2 Ground — unsuitable collocation of "aquifers" (Line 54)**

RC: **Remove "ground" (!?) and just keep "aquifers".**

RV: Yes; thank you for the suggestion. We have removed the word "ground" and used the term "aquifers" alone.

**2.4.3 Choice of standardization (Section 3.2)**

RC: **I understand the arguments supporting the choice of classical standardization instead of the usual minmax scaling. Yet, it would be interesting to indicate whether the two types of scaling were tested or not (from the text it seems that no trial was made using minmax scaling but it should be indicated).**

AR: No; we did not test the min max normalization.

RV: We have added the following sentence to the end of §3.2 of the (re-)revised manuscript: " We should, however, note that we have not tested other forms of normalization, for example, the min–max normalization ([0, 1] scaling), and have not investigated their influence on LSTM performance."
* * *
[1] term borrowed from Bengio (2012)

**2.4.4 Caption of Figure 9**

RC: **It seems to contradict the legend at the top of the figure (which says, for instance, "solid=mean" while the caption says "solid=training").**

AR: Thank you for noticing this mistake. The legend is explaining the correct correspondence.

RV: We have modified the caption of Figure 9 as follows: "[...] In each panel, the dashed and dotted lines correspond respectively to the training and validation data. The solid line is the mean of the training and validation lines. [...]"

**REFERENCES**

Y. Bengio. Practical recommendations for gradient-based training of deep architectures. In *Neural networks: Tricks of the trade*, pages 437–478. Springer, 2012.

I. Goodfellow, Y. Bengio, and A. Courville. *Deep Learning*. MIT Press, 2016.

D. P. Kingma and J. Ba. Adam: A method for stochastic optimization. *arXiv preprint arXiv:1412.6980*, 2014.

F. Kratzert, D. Klotz, C. Brenner, K. Schulz, and M. Herrnegger. Rainfall–runoff modelling using long short-term memory (lstm) networks. *Hydrology and Earth System Sciences*, 22(11):6005–6022, 2018.

F. Kratzert, D. Klotz, G. Shalev, G. Klambauer, S. Hochreiter, and G. Nearing. Towards learning universal, regional, and local hydrological behaviors via machine learning applied to large-sample datasets. *Hydrology and Earth System Sciences*, 23(12):5089–5110, 2019.

T. Lees, M. Buechel, B. Anderson, L. Slater, S. Reece, G. Coxon, and S. J. Dadson. Benchmarking data-driven rainfall–runoff models in great britain: a comparison of long short-term memory (lstm)-based models with four lumped conceptual models. *Hydrology and Earth System Sciences*, 25(10):5517–5534, 2021.